# Multi-ancestry GWAS of age-related hearing loss identifies 140 loci and key cellular mechanisms

Lulu Shi [1,3], Haibin He[1,3], Junpeng Li[1], Kai Gai[1], Wenjian Li[1], Yu Zhao[2], Huijun Yuan[1] ✉ & Yang Wu [1] ✉

Age-related hearing loss is a prevalent and growing public health issue among the elderly. Here, we perform a multi-ancestry genome-wide association study comprising 456,613 cases and 1,053,834 controls, identifying 140 independent loci associated with age-related hearing loss, including 44 novel signals. We further fine-map 9 likely causal missense variants for age-related hearing loss and provide evidence of purifying selection for age-related hearing loss-associated variants. Notably, genetic risk for age-related hearing loss is strongly correlated with behavior traits such as neuroticism score and irritability. Integration of molecular phenotypes identifies 22 genes and 85 DNA methylation sites significantly associated with age-related hearing loss. Moreover, analyses incorporating spatial and single-cell transcriptomic identify the inner ear as a crucial site of age-related hearing loss, emphasizing the importance of hair cells, supporting cells, basal and root cells of the stria vascularis to its pathogenesis. Our study provides genetic and cellular insights into age-related hearing loss and advance our understanding of its genetics architecture.

Age-related hearing loss (ARHL) is a type of sensorineural hearing loss that gradually develops with increasing age and is one of the most common health issues among the elderly. As the global aging populations continue to increase, the prevalence of ARHL is gradually increasing over years[1,2]. According to the World Health Organization (WHO), hearing loss affects over 25% of adults over the age 60, which severely impacts communications, social engagement and mental health[3–5]. Genetic factors are known to play a key role in ARHL, as evidenced by numerous family-based and population-level studies[6–8]. Genome-wide association studies (GWAS) have identified a growing number of loci associated with ARHL. For instance, the largest GWAS meta-analysis to date, performed by Trpchevska et al. in a European cohort of 723,266 individuals with ICD diagnoses and self-reported hearing loss, identified 48 associated loci and highlighted the biological relevance of the stria vascularis in ARHL pathogenesis[9].

Despite these advances, much of the genetic basis of ARHL remains unexplained. This is partly due to the high phenotypic and genetic heterogeneity of ARHL, as well as differences in phenotypic definition[10]. Therefore, larger sample sizes for ARHL GWAS are required to achieve statistical power comparable to that of other complex traits. Moreover, most existing GWAS have predominantly focused on individuals of European descent, which limits the generalizability of findings across diverse populations. Cross-ancestry analyses are therefore essential to improve our understanding of shared genetic architecture between ancestries and uncover ancestry-specific association signals. In addition, the increasing availability of high-resolution single-cell and multi-omics datasets enables integrative analyses that can link GWAS signals to specific genes, regulatory elements, and disease-relevant cell types.

In this study, we performed a large-scale cross-ancestry GWAS meta-analysis of ARHL, combining data from 456,613 cases and

[1]Department of Otolaryngology-Head and Neck Surgery & Institute of Rare Diseases, Frontiers Science Center for Disease-related Molecular Network, West China Hospital, Sichuan University, Chengdu, Sichuan, China. [2]Department of Otolaryngology-Head and Neck Surgery, West China Hospital, Sichuan University, Chengdu, Sichuan, China. [3]These authors contributed equally: Lulu Shi, Haibin He. ✉e-mail: yuanhj301@163.com; yang.wu@wchscu.edu.cn

1,053,834 controls from European (EUR), East Asian (EAS), African (AFR) and Admixed American (AMR) populations. We systematically explored the genetic architecture of ARHL, identified novel susceptibility loci, fine-mapped putative causal variants, prioritized functional genes and cell types through integration with multi-omics QTL datasets and spatially resolved transcriptomic data.

## Results

### Cross ancestry meta-analysis identifies 44 new risk loci for ARHL

An overview of the study is shown in Fig. 1. We performed a multi-ancestry GWAS meta-analysis by combining summary statistics from EUR, EAS, AFR and AMR populations ("Methods", Supplementary Fig. 1 and Supplementary Data 1). The primary analysis included 456,613

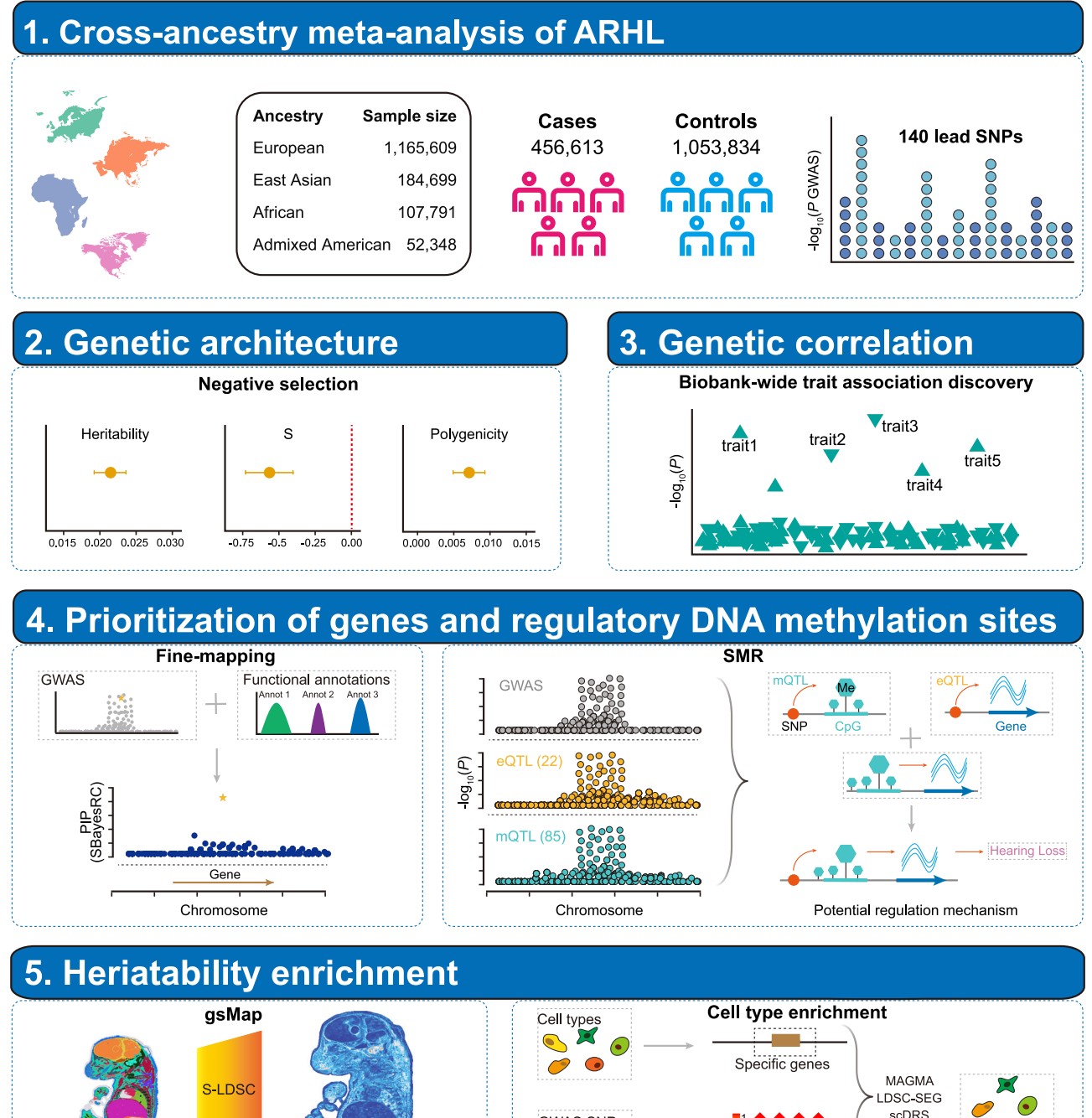

**Fig. 1 | Overview of this study.** The GWAS summary statistics from EUR, EAS, AFR and AMR were used for the cross-ancestry meta-analysis. Downstream analyses included estimation of genetic architecture, genetic correlation, fine-mapping, functional gene and regulatory DNA methylation prioritization, tissue and cell type heritability enrichment. ARHL age-related hearing loss, MAF minor allele frequency, SMR summary data-based Mendelian randomization, eQTL expression quantitative trait locus, mQTL methylation quantitative trait locus. The map of the continent in the top box was created in BioRender. Bu, F. (2026) https://BioRender.com/0ngxsjy.

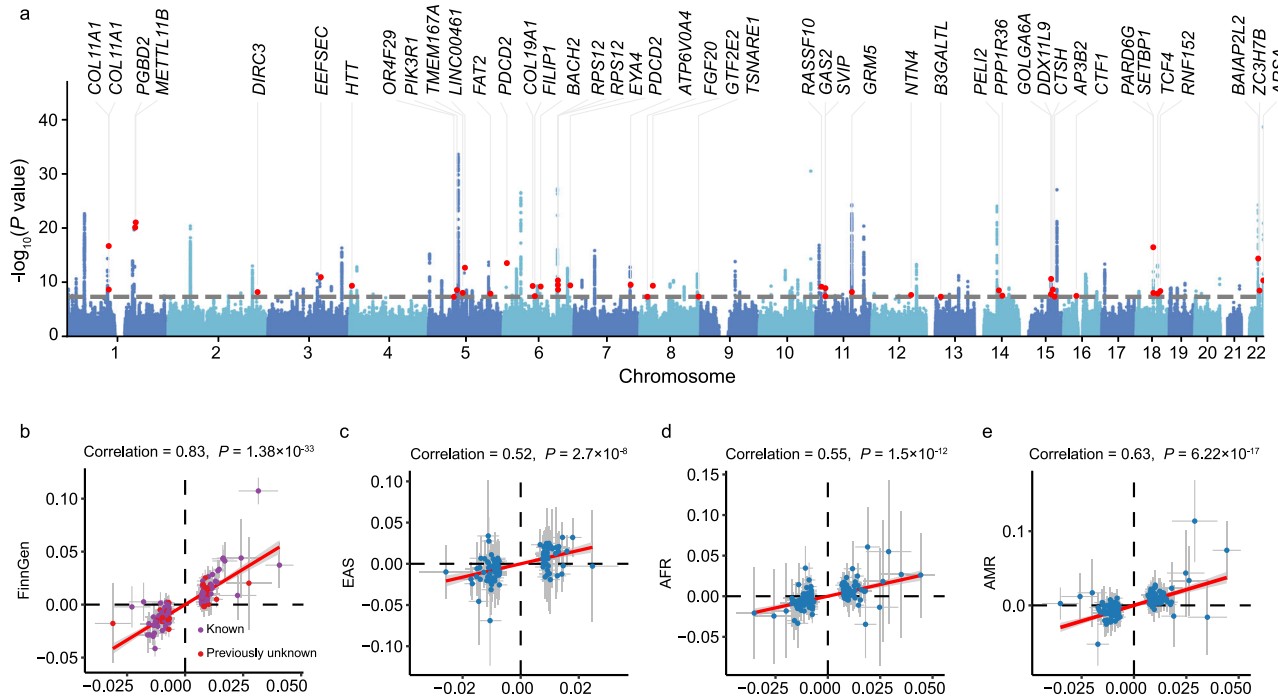

**Fig. 2 | Cross-ancestry GWAS meta-analysis for ARHL. a** Manhattan plot of the cross-ancestry GWAS meta-analysis. The horizontal gray line denotes the genome-wide significance threshold ($P < 5 \times 10^{-8}$). In total, 44 novel independent risk loci are highlighted in red, with their nearest annotated genes shown in black text. The $P$ values are based on a two-sided $t$ test. **b** Effect size comparison between the discovery GWAS meta-analysis and the independent replication dataset from FinnGen. Of the 140 lead SNPs, 127 were available in FinnGen. Novel SNPs are highlighted in red. The Pearson's coefficient for effect sizes is 0.83 ($P = 1.38 \times 10^{-33}$, two-sided $t$ test). Error bars represent 95% confidence intervals (CI) derived as effect size ±

1.96 × se. **c–e** Pairwise comparisons of lead SNP effect sizes between European and three other ancestry groups, East Asian ($m = 100$ SNPs available), African ($m = 140$ SNPs available) and Admixed American ($m = 140$ SNPs available). Each dot shows the effect size of a lead SNP, and the error bars show the 95% CI computed from the effect size estimates (β ± 1.96 × se). Effect size correlations were computed using Pearson's r (two-sided $t$ test for $P$-values). Each variant constitutes an independent data point. EUR European ancestry, EAS East Asian ancestry, AMR Admixed American ancestry, AFR African ancestry.

cases and 1,053,834 controls, identifying 140 independent loci associated with ARHL at genome-wide significance ($P < 5 \times 10^{-8}$), of which 44 were novel ("Methods"; Fig. 2a, Supplementary Fig. 2 and Supplementary Data 2). Regional association plots revealed that several of these novel loci reside in regions with complex linkage disequilibrium (LD) patterns, and the functional genes underlying these observed associations remain largely unresolved (Supplementary Fig. 3). To assess whether confounding factors such as population stratification exist, we applied LD score regression (LDSC)[11]. The LDSC regression intercept was 1.0305 (SE = 0.0104) together with λ1000 (1.0006), suggesting that the observed genomic inflation was attributable to polygenic architecture rather than confounding (Supplementary Fig. 2). The estimated SNP-based heritability ($h^2_{SNP}$) on the liability scale was 0.0219 (SE = 0.0009), assuming a population prevalence of 0.1 ("Methods"). To replicate the identified 140 lead SNPs, we used an independent cohort from FinnGen[12], comprising 39,620 cases and 397,865 controls. Of the 140 SNPs identified, 127 were available for replication analysis. Among them, 15 SNPs (11.81%) reached the genome-wide significance ($P < 5 \times 10^{-8}$), 37 SNPs (29.13%) reached the Bonferroni corrected significance ($P < 0.00039$), and 81 SNPs (63.78%) reached the nominal significance ($P < 0.05$). In addition to high replication rates, we observed high concordance in the effect sizes between the meta-analysis discovery and FinnGen replication cohort ($r = 0.83$, $P = 1.38 \times 10^{-33}$; Fig. 2b). Popcorn[13] analysis further estimated a genetic correlation of 0.63 (se = 0.12) between the two datasets, supporting a moderate but robust consistency in the underlying genetic architecture. For comparison with the largest previous ARHL GWAS by Trpchevska et al.[9], 102 of the 140 identified SNPs overlapped between datasets. Of these, 45 SNPs (44.12%) reached genome-wide significance

($P < 5 \times 10^{-8}$), and all of them reached the Bonferroni corrected significance ($P < 0.00049$). To further assess the cross-ancestry consistency, we compared effect size estimates and effect allele frequency (EAF) between the EUR population and each of the three non-EUR groups. The results showed moderately high cross-ancestry concordance in both effect sizes and EAF, indicating a largely shared genetic architecture of ARHL across diverse populations (Fig. 2c–e and Supplementary Fig. 4).

## Natural selection of ARHL-associated variants

To investigate evolutional constraints acting on identified ARHL-associated loci, we examined the relationship between minor allele frequency (MAF) and effect size. We observed a significant negative relationship, where variants with lower MAF tended to have larger effects on ARHL (Fig. 3a). This pattern is consistent with a negative selection model, where deleterious variants are kept at low frequencies in the population[14,15]. To formally assess this, we performed a SBayesS[16] analysis, a method implemented in GCTB, which quantifies the relationship between effect size variance and MAF using the $S$ parameter ("Methods"). Analyzing each chromosome separately, we observed a significantly negative genome-wide $S$ estimates ($\hat{S} = -0.52$, SE = 0.06), providing robust evidence for negative selection acting on rare deleterious variants (Fig. 3b). In addition to selection inference, SBayesS method also allowed us to estimate the $h^2_{SNP}$ and polygenicity (π, i.e., the proportion of SNPs with non-zero effect) jointly with the $S$ parameter on each chromosome. We found that the $h^2_{SNP}$ of each chromosome was strongly correlated with chromosome length ($r = 0.91$; Fig. 3c), indicating a broadly distributed polygenic architecture for ARHL. The mean polygenicity estimate across chromosomes (i.e., π)

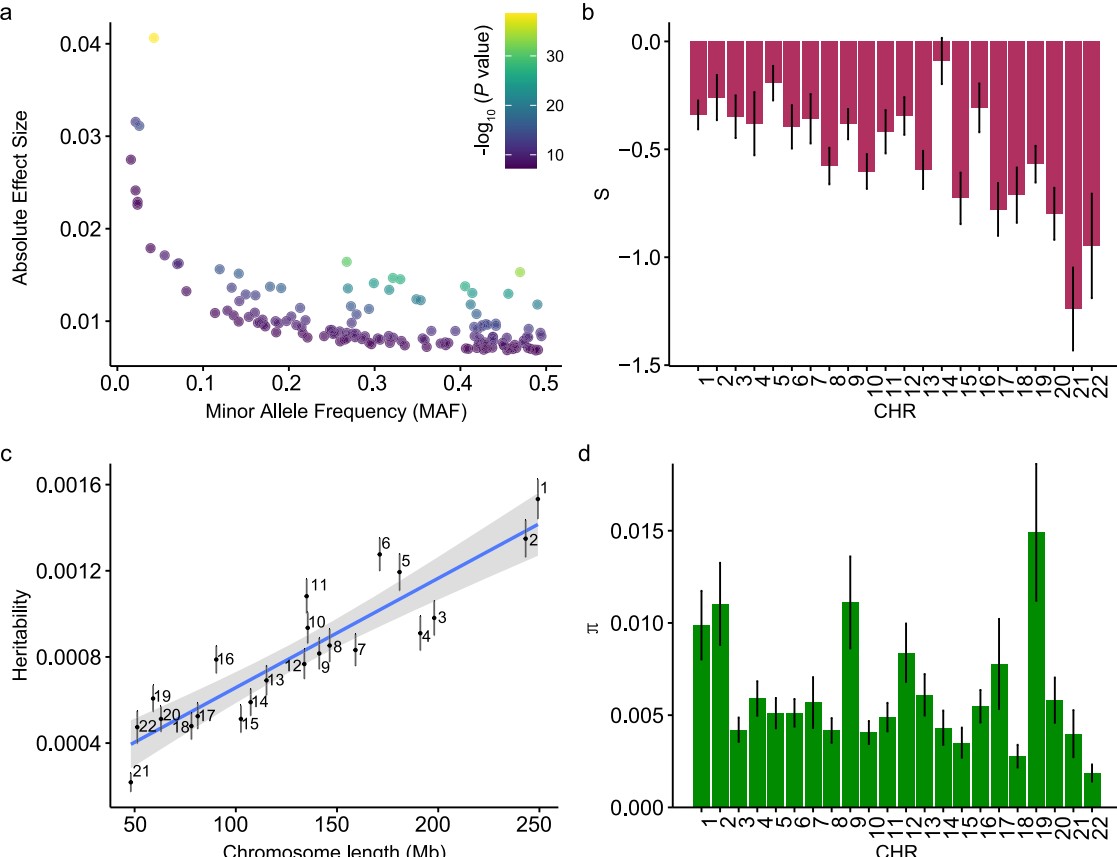

**Fig. 3 | Estimated genetic architecture of ARHL. a** Relationship between the effect sizes and minor allele frequency for genome-wide significant lead SNPs. Each point represents an SNP, colored by its significance level in -log$_{10}$(*P*-value) scale. The *P*-values are based on two-sided *t* test. **b** Chromosome-wide estimates of SBayesS selection parameter (*S*), where more negative values indicate stronger evidence of negative selection. Each bar represents one independent chromosome (*m* = 22), error bars denote the posterior standard deviation (posterior SD) obtained from the Markov Chain Monte Carlo (MCMC) samples of the SBayesS model. **c** Relationship between chromosome-wide SNP-based heritability and chromosome length. Each point corresponds to one chromosome (*n* = 22). The blue line represents the fitted linear regression, and the shaded area indicates the 95% CI of the regression estimate. Chromosome-wide heritability estimates and their uncertainty were obtained as posterior means ± posterior SD from the SBayesS MCMC output. **d** Estimates of proportion of SNPs with non-zero effects (π; i.e., polygenicity) for each chromosome (*n* = 22). Bars correspond to posterior means, with error bars representing posterior SD across MCMC iterations.

was 0.62%, suggesting that ARHL is influenced by a large number of variants with small effects (Fig. 3d). Taken together, these results demonstrate that ARHL is shaped by widespread negative selection and high polygenicity. The evidence of purifying selection further suggests that future whole-genome sequencing (WGS) studies in larger and more diverse cohorts are likely to uncover additional rare, high-impact variants that contribute to ARHL susceptibility.

**Shared genetic analyses highlight link between ARHL and behavior traits**

To investigate the genetic overlap between ARHL and other complex traits, we performed a genetic correlation analysis across 1738 phenotypes in the UK Biobank (UKB)[17]. To avoid the potential bias due to sample overlap, we applied de-meta-analysis to our GWAS summary statistics to exclude the UKB cohort before the analysis ("Methods", Supplementary Fig. 5). This yielded a dataset comprising 342,295 cases and 730,385 controls (total *N* = 1,072,680). We observed a high cross-population genetic correlation of 0.98 (se = 0.0026), indicating strong concordance in genetic architecture between the full meta-analysis and the de-meta-analysis results. Using BADGERS[18], a polygenic risk scores-based approach for biobank-wide association scans, we identified 121 traits significantly associated with ARHL after Bonferroni correction (*P* < 2.88 × 10$^{-5}$; Fig. 4a and Supplementary Data 3). As expected, hearing-related traits exhibited the strongest associations with ARHL (Fig. 4a). The most significant associations

were observed with the diagnosis of hearing difficulty, including hearing difficulty with background noise and general hearing loss (*P* < 1 × 10$^{-300}$), further validating the robustness of our GWAS results. In addition to auditory phenotypes, we observed a broad range of health and behavior-related phenotypes genetically linked to ARHL, suggesting its systemic and psychological effects. We then classified these 121 significantly associated traits into 10 categories based on their clinical or functional relevance (Supplementary Data 4). Among non-auditory categories, we observed that physical health and pain, as well as mental health and mood, showed stronger associations with ARHL than other categories. Notable examples include long-standing illness (*P* = 1.03 × 10$^{-21}$), chest pain or discomfort (*P* = 5.11 × 10$^{-16}$), neuroticism score (*P* = 1.33 × 10$^{-24}$) and irritability (*P* = 1.03 × 10$^{-16}$). In the social and family history category, loneliness (*P* = 7.52 × 10$^{-8}$) exhibited the strongest association with ARHL. Furthermore, we also observed strong correlations between ARHL and insomnia (*P* = 1.97 × 10$^{-11}$), frequency of depressed mood in last 2 weeks (*P* = 1.85 × 10$^{-13}$) and past tobacco smoking (*P* = 2.94 × 10$^{-6}$), consistent with previous findings[9]. To assess the robustness of genetic correlation findings using BADGERS with de-meta-analysis results, we selected 12 representative phenotypes based on either being the most significantly associated ARHL within each non-auditory category (as defined above) or representing previously established associations replicated in our study. The estimated genetic correlations between the ARHL meta-analysis and these

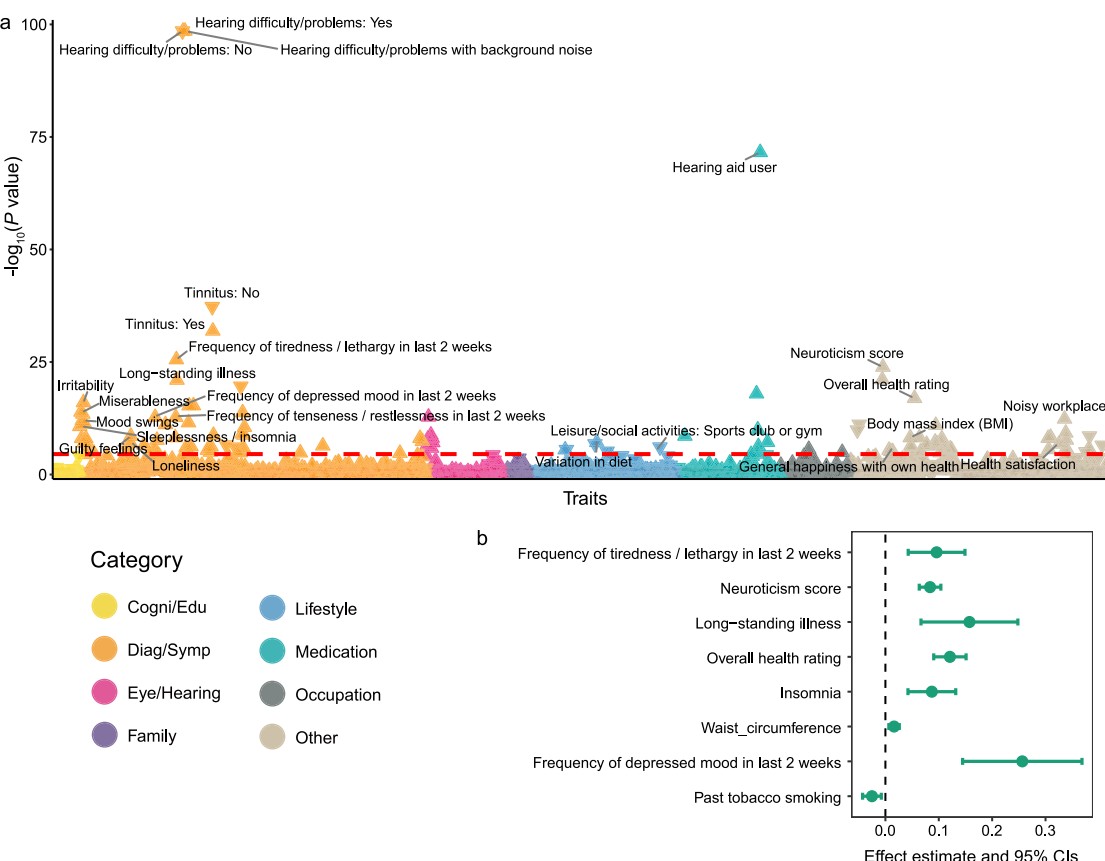

**Fig. 4 | Identification of putative risk factors for ARHL. a** A biobank-wide association scan was performed to identify an association between 1738 human heritable traits from the UK biobank and ARHL. Each triangle represents a trait, colored by different trait categories. The *P*-values are based on two-sided *t* test and truncated at $1 \times 10^{-100}$ for visualization. The red horizontal line denotes the significance threshold after Bonferroni correction ($P < 0.05/1738$, $2.88 \times 10^{-5}$). Upward-pointing triangles indicate positive associations, and downward-pointing triangles indicate negative associations. **b** Results of putative causal associations from Mendelian Randomization analysis using ARHL as outcome. Each point represents one exposure trait, with the estimated causal effect obtained from GSMR. Error bars indicate 95% CI computed as estimated causal effect ± 1.96 × se, where se is the standard error estimated from the GSMR.

phenotypes remained strong and statistically significant (Supplementary Data 5), further supporting for the reliability of the BADGERS findings.

Using the same set of 12 significant phenotypes, we applied GSMR[19,20], a multi-SNP-based Mendelian randomization (MR) method, to explore potential causal relationships between ARHL and these nonauditory traits ("Methods"). Of these, 8 traits showed significant evidence of causal association with ARHL at a false discovery rate threshold (FDR < 0.05; Fig. 4b and Supplementary Data 6). We further assessed the robustness of GSMR associations using five additional MR approaches ("Methods"). Across the 8 traits identified as significant by GSMR, the associations remained significant and showed consistent effect directions across all other methods, reinforcing the reliability of these associations (Supplementary Fig. 6 and Supplementary Data 7). These findings highlight the complex interplay between hearing loss and behavioral traits, suggesting potential shared biological pathways.

**Genome-wide fine mapping identifies 9 missense variants associated with ARHL**

To prioritize putative causal variants underlying ARHL-associated loci, we performed statistical fine-mapping using a state-of-art genome-wide fine-mapping (GWFM)[21] approach, which incorporates the functional annotation and genome-wide LD information. We used GWFM because it has demonstrated superior performance in identifying causal variants than the regional-based fine-mapping approaches[21]. Prior to fine-mapping, we first used the GWFM to assess heritability enrichment across 96 functional annotations defined in Gazal et al.[22].

The analysis highlighted the top 20 enriched categories, with the fold enrichment values ranging from 3.0 to 16.8 (Supplementary Fig. 7). Notably, the nonsynonymous annotation showed the highest per-SNP heritability enrichment. We further observed that nonsynonymous annotation was enriched not only for a higher proportion of putative causal variants but also for variants with larger effect sizes (Supplementary Fig. 8), underscoring its biological importance in ARHL pathogenesis. Applying a posterior inclusion probability (PIP) threshold of 0.9, we identified 1108 SNPs from 165 credible sets, collectively explaining 7.6% of the total $h^2_{SNP}$ (Supplementary Data 8). Among these, 22 SNPs showed strong causal evidence (PIP > 0.9) for ARHL, including 9 missense variants (Supplementary Fig. 9 and Supplementary Data 9, 10). Of these 9 missense variants, 4 are located in genes previously validated in hereditary hearing loss, according to the Hereditary Hearing Loss Homepage (https://hereditaryhearingloss.org/): *LOXHD1* (DFNB77, [MIM: 613072]), *CLRN2* (DFNB117, [MIM: 618988]), *TMPRSS3* (DFNB8/10, [MIM: 605511]) and *EYA4* (DFNA10, [MIM: 603550]). In addition, *KLHDC7B* and *COL9A3* have been linked to auditory function in mice, as reported by the International Mouse Phenotyping Consortium (IMPC, https://www.mousephenotype.org/about-impc/). To evaluate the robustness of our fine-mapping results, we re-analyzed all loci using the SuSiE[23,24] with the same LD reference. In the diagnostic analysis, SuSiE estimated a mean lambda of 0.006 across loci (Supplementary Fig. 10), indicating good concordance between the LD reference and our ARHL GWAS data. Further fine-mapping analysis with SuSiE identified 9 SNPs with PIP > 0.9, of which 7 overlapped with the 22 SNPs identified by our GWFM analysis. Among

the 22 fine-mapped variants identified by GWFM, 7 achieved a SuSiE PIP > 0.9, and 50% had PIPs > 0.5 and 87% had PIPs > 0.1 (Supplementary Data 11), confirming the consistency and reliability of our findings. These fine-mapping results highlight the contribution of missense variants to ARHL susceptibility and provide a refined set of high-confidence candidate variants and genes for future functional validation.

### Prioritization of functional genes and DNA methylations

To prioritize functional genes for these identified GWAS loci in non-coding regions, we performed summary data-based Mendelian randomization (SMR)[25] analysis using the top-associated expression quantitative trait locus (eQTL) as an instrument variable to test for associations between gene expression level of each gene and ARHL risk ("Methods"). We used the GWAS summary data from our meta-analysis and eQTL summary data from eQTLGen consortium[26] ($n = 14,115$) for the SMR analysis. In total, we identified 47 genes at a Bonferroni corrected significance level ($P_{SMR} < 3.20 \times 10^{-6}$). To filter out the SMR associations caused by linkage (i.e., the causal variants for gene expression and the causal variants affecting ARHL risk were in linkage disequilibrium), we further performed the Heterogeneity Independent Instruments (HEIDI) analysis for all significant SMR associations (Methods). Genes not rejected by the HEIDI test ($P_{HEIDI} > 0.01$) are likely influenced by shared causal variants affecting both gene expression and ARHL. Of the 47 genes that passed the SMR test, 22 were not rejected by the HEIDI test ($P_{HEIDI} > 0.01$; Supplementary Fig. 11 and Supplementary Data 12), implying a pleiotropic association. To further prioritize the regulatory DNA methylation (DNAm) sites associated with ARHL risk, we performed the SMR and HEIDI analyses using methylation quantitative trait locus (mQTL) data from Brisbane Systems Genetics Study and Lothian Birth Cohorts[27] ($n = 1980$) to identify DNAm sites associated with ARHL through pleiotropy at a shared causal variant. In total, we identified 85 DNAm sites associated with ARHL at a genome-wide significance level ($P_{SMR} < 5.37 \times 10^{-7}$) and $P_{HEIDI} > 0.01$ (Supplementary Fig. 12 and Supplementary Data 13).

To infer the plausible genetic regulatory mechanisms, we integrated these 85 ARHL-associated DNAm sites with 22 ARHL-associated genes identified through the SMR and HEIDI test above. We identified 44 DNAm site-gene pairs showing pleiotropic associations across 9 genes ($P_{SMR} < 5.38 \times 10^{-4}$, $P_{HEIDI} > 0.01$; Supplementary Data 14). The identification of these putative functional genes and their regulatory elements provides opportunities to infer the mechanism of genetic regulation at a GWAS locus. A notable example is the 17p13.1 locus, where three genes (*ACADVL*, *DVL2* and *ELP5*) showed consistent association signals across data from GWAS, eQTL and mQTL studies, implying a plausible biological pathway. In this region, the ARHL-associated DNAm probes cg10691912, cg00072720, cg12805420 and cg11941546 were simultaneously associated with the expression level of all three genes, suggesting a potential co-regulation via DNAm (Supplementary Fig. 13). Among these three genes, *ACADVL* emerged as a strong candidate gene with the following evidence. First, the expression levels of *ACADVL* were positively associated with ARHL risk. Second, the SNP-association signals were identified for the DNAm probe cg19466160, and the methylation level of this probe was significantly associated with gene *ACADVL* (Fig. 5 and Supplementary Data 13 and 14). Third, the DNAm probe cg19466160 is located near the enhancer regions of gene *ACADVL* according to the chromatin state annotations from the Roadmap Epigenomics Mapping Consortium (REMC) reference samples[28]. Together, these findings support a plausible regulatory mechanism in which the genetic variant alters the DNAm level at cg19466160, which in turn down-regulates the expression of the *ACADVL* gene and therefore decreasing the risk of ARHL.

### Tissue and cell type-specific enrichment of ARHL heritability

To identify biologically relevant tissue for ARHL, we applied gsMap[29] method to quantify the spatial tissue and cell type-specific SNP heritability enrichment ("Methods"). Due to the lack of publicly available spatial transcriptomics (ST) data from human embryos, we used mouse embryo ST data from MOSTA[30] database as a proxy (Fig. 6a and Supplementary Data 15). As expected, the inner ear region showed the strongest heritability enrichment for ARHL (Cauchy-combination *P*-value = $3.97 \times 10^{-6}$, Fig. 6b and Supplementary Data 16), validating its central role in ARHL etiology.

To refine these tissue-level findings at the cellular level, we leveraged the single-cell RNA-seq (scRNA-seq) data from the mouse cochlear by Jean et al.[31], which profiled 85,167 cells across 36 distinct cochlear cell types. Using this dataset, we applied three complementary approaches (i.e., LDSC-SEG, MAGMA and scDRS) to evaluate SNP heritability enrichment and to quantify disease relevance across cochlear cell types[32–34] (Methods). Across these analyses, hair cells and supporting cells consistently showed the strongest enrichment, suggesting they are the primary contributors to the genetic architecture of ARHL (Fig. 6c and Supplementary Data 17). Moreover, significant enrichment signals were also observed in the basal and root cell compartments of the stria vascularis in both MAGMA and scDRS analyses, highlighting a potential complementary role of these non-sensory epithelial cells in hearing loss pathogenesis. To further validate these results, we repeated the analysis with four additional mouse cochlear scRNA-seq datasets used by Eshel et al.[35–38], which spanned a broad age range from postnatal day 15 (P15) to 15 months. Similar enrichment patterns were observed in these datasets (Supplementary Fig. 14 and Supplementary Data 18). These findings reconcile the apparent discrepancy between Eshel et al.[37], who emphasized hair cells and supporting cells, and Trpchevska et al.[9], who highlighted basal and root cells of the stria vascularis, by revealing their shared contribution to hearing loss. Collectively, these results provide robust evidence that ARHL heritability is significantly enriched in hair cells, supporting cells, and basal and root cells of the stria vascularis, reinforcing their collective roles in hearing loss pathogenesis.

## Discussion

In this study, we conducted the largest cross-ancestry GWAS meta-analysis of ARHL to date, incorporating data from 1,510,447 individuals (456,613 cases and 1,053,834 controls) of EUR, EAS, AFR and AMR ancestry. We identified 140 independent loci associated with ARHL ($P < 5 \times 10^{-8}$), including 44 novel loci not reported in previous studies. Through statistical fine-mapping, we prioritized 9 likely causal missense variants. By integrating multi-omics QTL data, we further identified 22 functionally relevant genes and 85 DNA methylation sites associated with ARHL. Moreover, by leveraging spatial and single-cell transcriptomic data, we highlighted the importance of hair cells and supporting cells to the pathogenesis of hearing loss. Compared with the study by Trpchevska et al.[9], our work provides three key advances. First, whereas Trpchevska et al. focused only on European cohorts, we performed a cross-ancestry meta-analysis, enabling the assessment of cross-population similarities and improving the generalizability of findings. Second, by increasing the sample size from ~0.7 million to ~1.5 million individuals, we substantially boosted discovery power and identified 140 independent loci, nearly a threefold increase in locus discovery. Third, our cell type enrichment analyses highlighted hair cells and supporting cells as the most strongly enriched cell types, while also revealing significant enrichment in basal and root cells of the stria vascularis, thereby reconciling the findings of Eshel et al.[37] and Trpchevska et al.[9]. The apparent discrepancy between the two studies likely reflects differences in cell-type coverage and dataset composition. Our findings deepen the understanding of the genetic

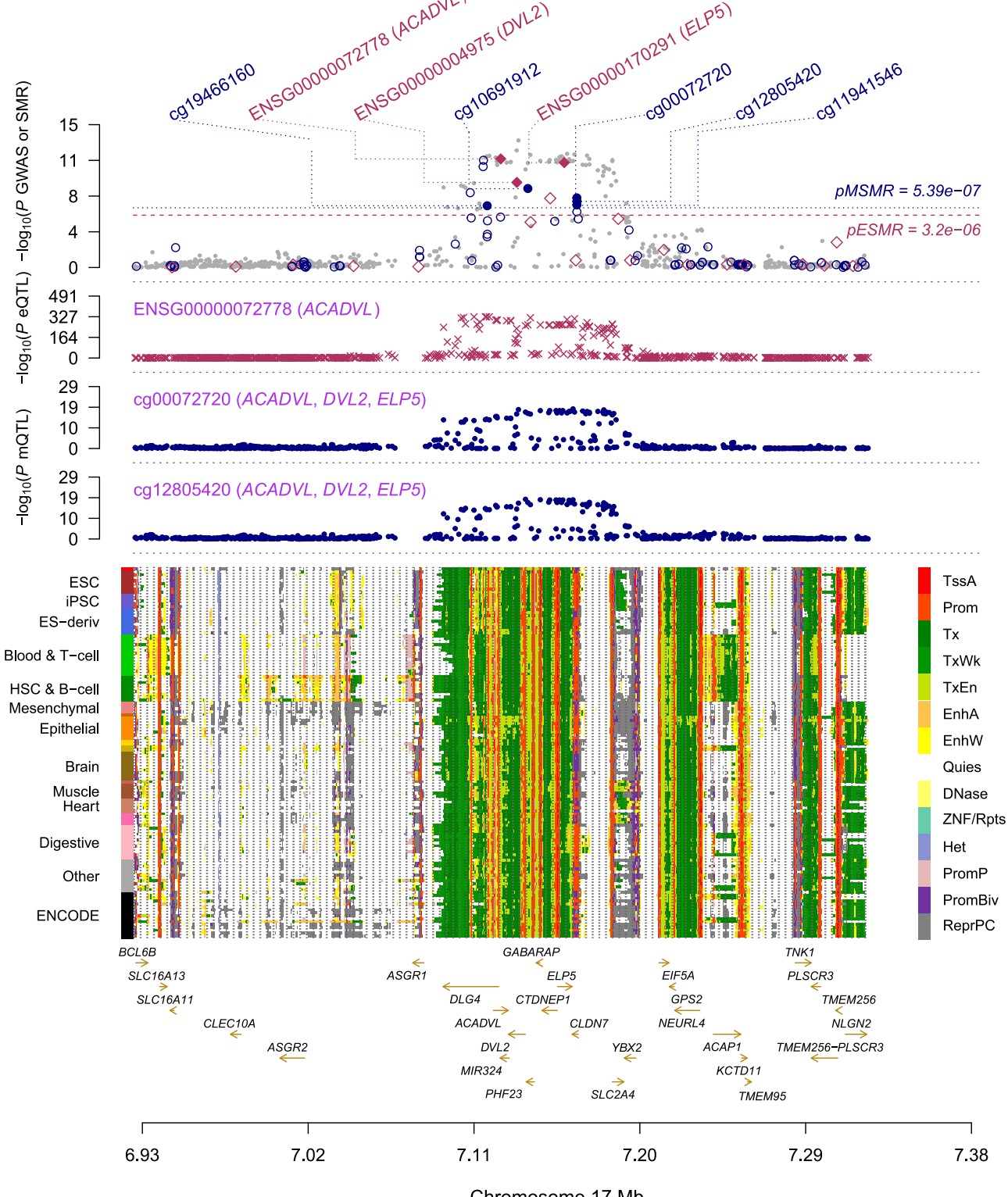

**Fig. 5 | Prioritizing genes and regulatory DNA methylation sites at the *ACADVL* locus for ARHL.** Results of SMR analysis that integrates GWAS summary statistics with the mQTL and eQTL studies are shown. The top plot shows -log₁₀(*P*-value) of SNPs from the GWAS meta-analysis for ARHL. The red diamonds and blue circles represent -log₁₀(*P*-value) from the SMR tests for associations of eQTL and mQTL probes with AHRL, respectively. Solid diamonds and circles represent the probes not rejected by the HEIDI test. The second plot (red diamonds) shows -log₁₀(*P*-value) of the SNP association for eQTL probe ENSG00000072778 (tagging *ACADVL*). The third plot (blue circles) shows -log₁₀(*P*-value) of the SNP association with mQTL probes cg00072720 and cg12805420. All *P* values are based on a two-sided *t* test. The bottom plot shows 14 chromatin state annotations of 127 samples from REMC for different primary cells and tissue types.

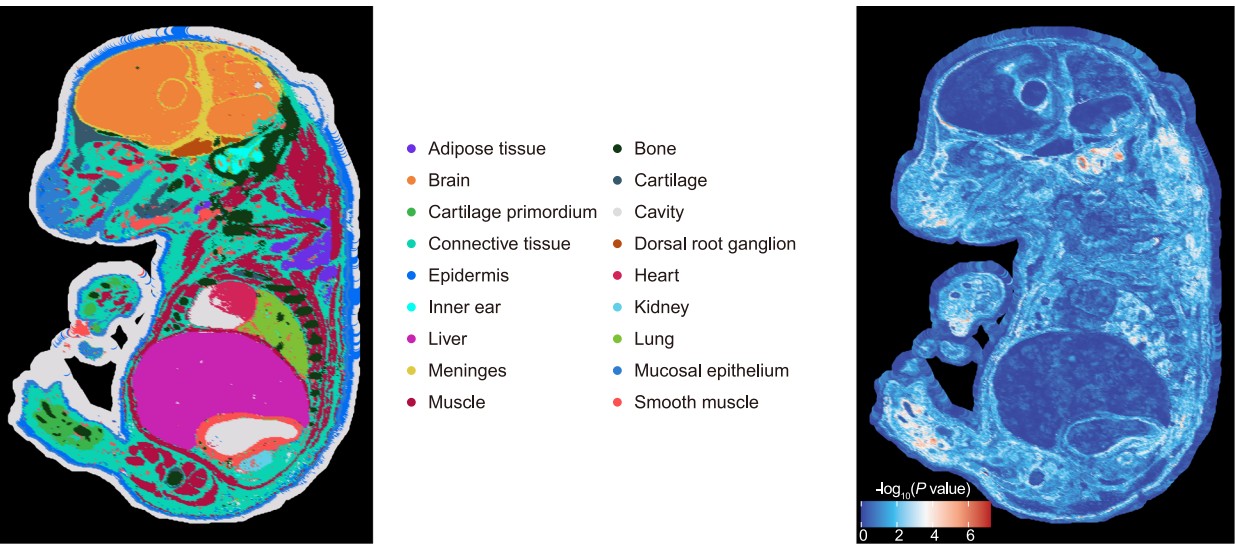

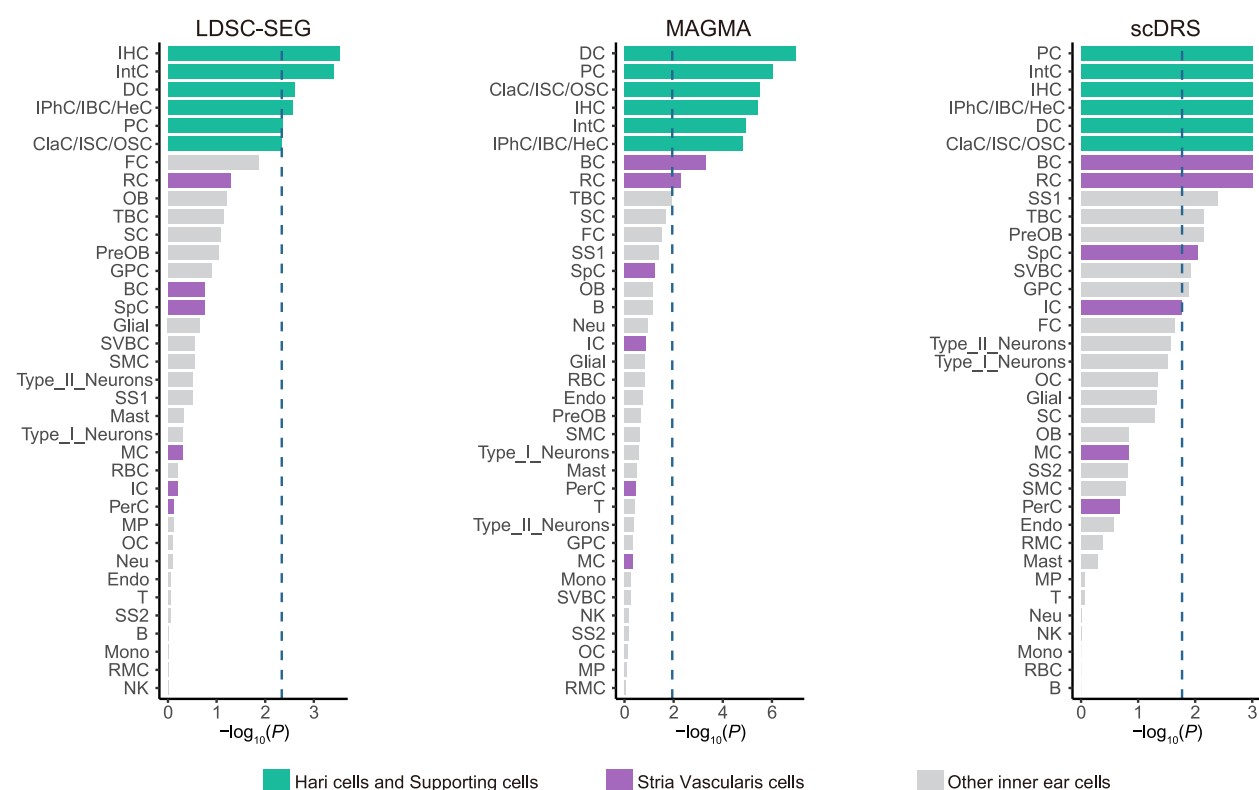

**Fig. 6 | Genetically informed mapping of ARHL-associated SNPs to spatial and single-cell transcriptomic data. a** Mouse E16.5 embryonic spatial transcriptomic (ST) data at bin50 resolution, with spots color-coded by 18 tissue types. **b** Spatial heritability enrichment results of ARHL from gsMap that integrate GWAS summary statistics with ST data. Colors indicate the strength of statistical significance of spot-trait associations. *P*-values are based on a two-sided *t* test. **c** Cell type-specific heritability enrichment across three methods using mouse cochlear single cell RNA-seq data. Each column corresponds to one method. Significance was controlled at FDR < 0.05 using the Benjamini-Hochberg (BH) procedure. The dashed line shows -log₁₀(*P*), where *P* is the dataset specific nominal threshold (raw *P* of the least BH-significant test); if absent, no tests passed BH-FDR < 0.05. *P*-values are based on a two-sided *t* test. B B cell, BC basal cell, ClaC/ISC/OSC Claudius/Inner/Outer sulcus cell, DC Deiters' cell, Endo Endothelial cell, FC fibrocyte, GPC Glial precursor cell, IPhC/IBC/HeC Inner border/phalangeal/Hensen cell, IHC Inner hair cell, IntC Inter-dental cell, IC Intermediate cell, IHC inner hair cell, MP macrophage, MC marginal cell, MC Mast cell, Mono Monocytes, Neu Neutrophils, OB Osteoblasts PerC Peri-cytes, PreOB Pre osteoblasts, RMC Reissner membrane cell, RC Root cell, SVBC Scala vestibuli border cell, SC Schwann cell, SMC Smooth muscle cell, SpC Spindle cell, RBC red blood cells, T T cell, TBC tympanic border cell.

architecture and regulatory mechanisms of ARHL, linking associated genetic variants to cell types and molecular phenotypes.

Our analysis highlights several important features of ARHL's genetic architecture. First, we found evidence of negative (purifying) selection, with rare variants of larger effects on ARHL are subject to evolutionary constraint. Second, ARHL showed high polygenicity, implying that many genetic variants contribute to disease risk, and that additional loci are likely to be discovered with larger and more diverse cohorts. This is further supported by extensive genetic correlations with other complex traits, where 121 of 1738 heritable traits showed significant associations with ARHL. These associated traits span diverse categories, including hearing-related phenotypes, clinical diagnoses, symptoms, medication usage, lifestyle factors and behavior traits, highlighting the interplay of both genetic and environmental contributors to ARHL.

Functionally, many of our prioritized genes have strong biological relevance to ARHL. Among the 9 fine-mapped missense variants, we replicated 3 genes (*EYA4*, *KLHDC7B* and *COL9A3*) previously reported by Trpchevska et al.[9]. We also identified 3 additional genes (*CLRN2*, *LOXHD1* and *TMPRSS3*) that are well-established causes of autosomal-recessive non-syndromic hearing loss[39–41]. For example, *CLRN2*, which encodes a tetraspan protein, has been shown to be essential for the long-term hearing maintenance in mouse models[42]. While certain genes are primarily implicated in monogenic non-syndromic hearing loss, their reduced penetrance in the general population likely reflects the polygenic architecture of age-related hearing loss, as well as the influence of genetic modifiers and environmental factors. These genes both provide independent support for genes previously implicated in familial hearing loss and highlight shared biological mechanisms between monogenic and polygenic forms of hearing loss. In our integration analysis of QTL data, we further highlighted several genes with a possible regulatory relationship linking genetic variants to DNAm, gene expression, and ARHL risk. At the 17p13.1 locus, for instance, we prioritized three genes (*ACADVL*, *DVL2* and *ELP5*) with regulatory DNAm associated with hearing loss. Notably, *DVL2* has been implicated in planar cell polarity (PCP) proteins of the inner ear sensory epithelia[43,44], which is essential for inner ear development.

Our spatial and single-cell transcriptomic analyses provide further insight into the cell-type specificity of ARHL. Given the current lack of adult human spatial transcriptomic (ST) datasets, a surrogate model is necessary. A recent study used mouse embryonic ST data revealed the male pattern baldness (MPB) risk gene enrichment in facial epithelial cells that spatially cluster during the formation of hair follicles and express canonical follicle marker genes (*KRT15*, *KRT5*, and *KRT17*)[29]. This example illustrates that embryonic ST data can provide valuable insights into the developmental origins of late-onset diseases. Using ST data of the mouse, we observed that ARHL-associated signals were mostly enriched in the inner ear (Fig. 6b). To resolve cell-type-specific contributions, we analyzed five mouse cochlear scRNA-seq data, identifying strong enrichment in hair cells and support cells and additional signals in basal and root cells of the stria vascularis. These findings reconcile the previously conflicting results of Eshel et al.[37] and Trpchevska et al.[9], providing a unified view of the cellular basis of ARHL heritability. Importantly, our study leveraged GWAS datasets with exactly twice the sample size of Trpchevska et al[9], thereby providing greater statistical power and confidence in the observed enrichments. In the work of Trpchevska et al., two mouse cochlear scRNA-seq datasets were analyzed, one captured only cells from the spiral ganglion region and the lateral wall/stria vascularis, while another was limited to inner hair cells (IHCs), outer hair cells (OHCs) and Deiters' cells[45,46]. We repeated heritability enrichment analysis using these two scRNA-seq datasets used in Trpchevska et al., and successfully replicated the reported enrichment in the stria vascularis by Trpchevska et al. (Supplementary Fig. 15 and Supplementary Data 19). When large-scale datasets with broader cell-type coverage were analyzed,

enrichment was observed not only in basal and root cells of the stria vascularis but also in hair cells and supporting cells, indicating that the differences between studies likely result from variation in dataset composition rather than conflicting biological interpretations (Fig. 6c). Specifically, the cell-type specificity scores of gene is related to a given cell type relative to all other cell types profiled. When the coverage of profiled cell types is limited, specificity scores can become inflated, leading to spurious enrichment. Taken together, our findings underscore the importance of using datasets with comprehensive cell-type coverage to achieve reliable inference of cell-type-trait associations.

We note several limitations of our study. First, although our study includes individuals from multiple ancestries, participants of EUR ancestry still accounted for the majority (~77%) of our GWAS samples. Given this predominant European contribution, and the lack of large-scale cross-ancestry replication cohorts and ancestry-specific LD reference panels, we replicated our findings in FinnGen and used the 1000 Genomes EUR samples as an LD reference for downstream analyses (e.g., SMR, LDSC). Although this strategy is consistent with those commonly used in cross-ancestry GWAS for other complex traits[47,48], it may introduce biases and constrain the generalizability of these results across populations[49]. Second, the heterogeneity in phenotypic definitions of ARHL across cohorts may introduce noise and limit statistical power. Future studies with a more explicit and standardized definition of ARHL, along with refined phenotyping methods, are needed. Third, previous study[50] suggests that ARHL may be influenced by sex-specific effects. However, as the majority of the used GWAS datasets lack results for sex-stratified analysis, we are unable to perform comprehensive sex-stratified meta-analyses. Moreover, the exclusion of the X chromosome from most GWAS datasets further limited our ability to capture X-linked or dosage-related effects. Taken together, these limitations highlight the need for future large-scale studies that incorporate both sex-stratified analyses and X-chromosome data to better characterize sex-specific contributions to ARHL. Fourth, the lack of transcriptomic data from human cochlear tissue and cell types remains a major limitation for hearing-related research. Although blood-based eQTL and mQTL can provide exploratory insights, the absence of cochlea-specific resources limits the ability to link the GWAS signal to the specific regulatory mechanisms. Similarly, while the mouse spatial and single-cell transcriptomic data are valuable, the potential interspecies differences and developmental-adult temporal mismatch may limit the interpretability of findings. Generating multi-omics QTL datasets, as well as spatial and single cell transcriptomic resources directly from adult human cochlear, will be essential to fully interpret GWAS findings. Fifth, some of the mouse scRNA-seq data used in our study were generated with the 10X Genomics platform, whose relatively shallow sequencing depth may have limited the number of detectable genes and rare cell subtypes, thereby reducing the ability to fully capture enrichment signals. As discussed above, the extent of cell-type coverage in the reference dataset critically influences the outcomes of heritability enrichment analyses. Accordingly, applying newer sequencing technologies that can capture more genes and resolve finer subtypes of cochlear cells will be essential to obtain more accurate and biologically meaningful insights. Finally, although we collected almost all available GWAS summary statistics for ARHL, the dataset from Ivarsdottir et al.[51] is currently missing from the Decode database, and that from Praveen et al.[52] is not publicly accessible. This lack of data accessibility limited our ability to perform a fully comprehensive meta-analysis, highlighting the need for broader data sharing to accelerate progress in the field.

## Methods

### Ethics

This study was approved by the Medical Ethical Committee of West China Hospital, Sichuan University [approval number: 2021 Audit (190)] and conducted in compliance with the Declaration of Helsinki.

The meta-analysis was performed using only publicly available or previously published GWAS summary statistics and involved no new recruitment or handling of individual-level data. Ethics approval and informed consent were obtained for all original GWAS studies by their respective institutional review boards or ethics committees.

## Data used in this study

The GWAS summary statistic data for ARHL analyzed in this study were obtained from four primary resources: the GWAS meta-analysis by Trpchevska et al.[9], the GWAS meta-analysis by De Angelis et al., the VA Million Veteran Program Consortium (MVP)[53] and Biobank Japan (BBJ)[54]. The meta-analysis dataset from Trpchevska et al.[9] was the largest ARHL GWAS meta-analysis in European ancestry to date, comprising 147,997 cases and 575,269 controls from 17 population-based cohorts. The meta-analysis dataset from De Angelis et al. leveraging the UK Biobank (UKB), the Nurses' Health Studies (I and II) and the Health Professionals Follow-up Study. Given that the GWAS summary statistics from Trpchevska et al.[9] already incorporated UKB participants, we thus removed UKB-derived signals from the De Angelis et al. dataset through de-meta-analysis prior to downstream analyses. The de-meta-analysis was performed using the following formulas:

$$SE_{\hat{\beta}_{DE-META}} = \sqrt{\frac{1}{\frac{1}{SE^2_{\hat{\beta}_{META}}} - \sum_c \frac{1}{SE^2_{\hat{\beta}_c}}}} \quad (1)$$

$$\hat{\beta}_{DE-META} = \hat{\beta}_{META} - SE^2_{\hat{\beta}_{DE-META}} \sum_c \frac{\hat{\beta}_c - \hat{\beta}_{DE-META}}{SE^2_{\hat{\beta}_c}} \quad (2)$$

where *META*, *c* and *DE-META* refers to the GWAS summary statistic from cross-ancestry meta-analysis, the cohort to be excluded (i.e., UKB) and de-meta-analysis, respectively. The P-value for each variant after de-meta-analysis was obtained using $2\Phi(-|\frac{\hat{\beta}_{DE-META}}{SE_{\hat{\beta}_{DE-META}}}|)$. After this adjustment, the dataset comprised 18,018 cases and 18,024 controls. The MVP GWAS summary data are ancestry-stratified, including EUR (229,580 cases and 176,720 controls), AMR (23,888 cases and 28,460 controls), EAS (2671 cases and 3302 controls) and AFR (31,058 cases and 76,733 controls). The GWAS summary data from BBJ included 3400 cases and 175,326 controls of East Asian ancestry. For replication, we used the GWAS summary statistics from the FinnGen[12], which includes 39,620 cases and 397,865 controls of EUR descent. The ARHL phenotype was defined either by ICD9/10 diagnoses of hearing loss (as applied in FinnGen, BBJ and MVP) or by self-reported hearing loss (De Angelis's GWAS and all cohorts in Trpchevska's GWAS except FinnGen). Self-reported hearing loss in included given prior evidence demonstrating that serves as a reliable proxy for formal hearing assessment[55]. Comprehensive details on phenotype definitions, sequencing platforms, quality-control and filtering criteria, case-control sample sizes and the analytical approaches used for GWAS analysis are provided in Supplementary Data 1.

## Cross-ancestry GWAS meta-analysis for ARHL

Prior to meta-analysis, we conducted standard quality control on all GWAS summary statistics. All datasets were harmonized to the hg19 reference genome build to ensure consistency of genomic coordinates and alleles. Duplicated SNPs and variants with inconsistent or mismatched alleles across the cohort were excluded. We further filter out SNPs with a minor allele frequency (MAF) < 0.01. We then conducted a two-stage meta-analysis strategy to combine GWAS summary statistics. In the first stage, we performed ancestry-specific meta-analyses within EUR (totaling 395,596 cases and 770,013 controls) and EAS (totaling 6071 cases and 178,628 controls). In the second stage, we performed a multi-ancestry meta-analysis by combining results from

EUR, EAS, AMR and AFR. This leads to a final GWAS summary statistic including 1,510,447 individuals for downstream analysis. All meta-analyses were performed using a fixed-effect inverse-variance-weighted (IVW) model implemented in METAL (the version released on 2011-03-25)[56]. The meta-analysis performed by METAL is based on the reported effect size and standard error for each SNP. Heterogeneity across studies was assessed using Cochran's Q test and the I² statistic.

To assess potential inflation due to population stratification or other confounding factors, we applied univariate LDSC[11]. The LDSC regression intercept would be close to one when the observed test statistic inflation is attributable to polygenicity rather than bias. In addition, we calculated λ1000 to further assess inflation standardized to a sample size of 1000 cases and 1000 controls. SNP-based heritability was then estimated on the liability scale using LDSC, assuming a population prevalence of 10% for ARHL. The genotype data of 1000 Genome Phase 3[57] European population was used as the LD reference.

To identify the independent genome-wide significant loci, we performed LD clumping using PLINK v1.90[58] with setting clumping r² 0.05, genome-wide significant P-value threshold as $5 \times 10^{-8}$ and the window size as 1 Mb around each index SNP. A locus was considered novel if it satisfied either of the following criteria relative to previously reported lead variants from large-scale ARHL GWAS (i.e., the variants reported by Ivarsdottir et al., Kalra et al., Praveen et al., De Angelis et al., Wells et al. and Trpchevska et al.[9,50,51,52,59]): it located > 500 kb away from any previously reported variants with $r^2 < 0.1$; or if it was within 500 kb of known loci but met a stricter criterion of LD independence ($r^2 < 0.05$).

## Estimating the genetic architecture for ARHL

To investigate the genetic architecture of ARHL, we first examined the relationship between MAF and the effect size of our genome-wide significant lead SNPs. A negative correlation between MAF and effect size may indicate the action of purifying selection on deleterious variants. To obtain a more comprehensive insight into the genetic architecture of ARHL, we applied the SBayesS[16] model, implemented in GCTB software, which estimates the joint distribution of SNP effect size and MAF, enabling inference of parameters of polygenicity, SNP-based heritability and the selection coefficient (*S*). For SBayesS analysis, we used the precomputed eigen-decomposition data provided by GCTB for genotype imputation and quality control. We further incorporated functional annotation information from Gazal et al.[22] and shrunk sparse LD matrices to improve the estimation of polygenicity and selection. Analyses were conducted at the chromosome level, allowing for comparison of genetic architecture across the genome.

## Genetic correlation analysis with other traits

To explore the shared genetic architecture between ARHL and a broad range of complex traits, we applied BADGERS[18], a PRS-based framework for biobank-wide association analysis. BADGERS estimates trait associations by leveraging GWAS summary statistics and a reference panel of PRS weights across multiple phenotypes. According to BADGERS, the reference panel comprises 1738 traits from the UKB[17], selected on the basis of nominally significant SNP-based heritability ($P < 0.05$). This selection ensures that only traits with sufficient heritability were retained for analysis. In this analysis, we evaluated the associations between ARHL and 1738 heritable traits from the UKB[17], which served as the reference trait panel for polygenic weights and covariance structure. To avoid potential inflation due to sample overlap—since our primary ARHL GWAS includes UK Biobank participants—we first removed UKB-derived signals from GWAS summary statistics through de-meta-analysis. We further assessed the concordance of genetic architecture between GWAS summary statistics from our meta-analysis and de-meta-analysis using Popcorn[13]. Finally, these de-meta-analyzed summary statistics were then used as input for BADGERS.

We further employed GSMR[19,20] method to investigate the potential causal associations between ARHL and 12 non-auditory phenotypes prioritized by BADGERS. GSMR uses generalized least squares to estimate causal effects, allowing multiple genome-wide significant SNPs as instrumental variables, thereby increasing statistical power and robustness against weak instrument bias. The genetic instrument variants were selected based on a clumping analysis using GSMR (i.e., $P = 5 \times 10^{-8}$ and LD $r^2 < 0.05$ within a 1 Mb windows). We used the 1000 Genomes Phase3 EUR population as the LD reference data for GSMR analysis. To further control for horizontal pleiotropy, we applied the HEIDI-outlier test (Heterogeneity In Dependent Instruments) with a significance threshold of $P > 0.01$ to exclude SNPs with potential horizontal pleiotropic effects from the analysis. We further validate the robustness of GSMR findings by applying five additional Mendelian randomization (MR) methods: inverse-variance weighting (IVW), pleiotropy residual sum and outlier (PRESSO), MR-Robust, Simple Median and robust adjusted profile score (RAPS)[60–63].

## Genome-wide fine mapping analysis

We performed genome-wide fine-mapping (GWFM) analysis using a Bayesian hierarchical mixture model implemented in GCTB[21]. In contrast to traditional region-based fine-mapping approaches (e.g., FINE-MAP), GWFM leverages the SBayesRC framework[64], a hierarchical multi-component mixture model that jointly integrates the genome-wide functional annotations and LD to fine-map the likely causal variants. To capture genome-wide LD patterns, we used the precomputed eigen-decomposition LD reference data provided by GCTB. Functional annotations were incorporated from the comprehensive dataset published by Gazal et al.[22], which includes 96 genomic features (e.g., coding, conserved region) relevant to complex trait architecture. Fine-mapping was performed across the entire genome, and variants with a posterior inclusion probability (PIP) $\geq 0.9$ were considered as likely causal. Additionally, credible sets were defined based on ranked PIPs to identify minimal SNP subsets accounting for 90% of the posterior probability at each signal. In addition, we applied the Sum of Single Effects (SuSiE)[23,24] framework to evaluate the robustness of GWFM findings. By employing the same LD reference, we ensured methodological consistency across analyses, allowing us to directly assess the concordance and stability of the identified signals.

## Summary-data-based Mendelian randomization analysis

To identify gene expression or DNA methylation levels associated with ARHL, we performed SMR and HEIDI[25] analyses, which integrate GWAS summary statistics for ARHL with molecular QTL data to distinguish pleiotropic associations from those driven by linkage[25]. We first performed SMR analysis to test the association between gene expression level and ARHL, using the top cis-eQTL SNP (i.e., within 2 Mb of gene expression probe and $P_{eQTL} < 5 \times 10^{-8}$) of gene as an instrument variable. The significant SMR associations can be caused by linkage, where two variants that one affecting the complex trait and another affecting gene expression level were in LD. We therefore applied the HEIDI test to distinguish pleiotropy (e.g., a causal variant affecting both trait and gene expression level) from the linkage using multiple SNP in the cis region. Similarly, we applied the SMR and HEIDI analyses to test for the association between DNAm and ARHL, and between ARHL-associated genes and ARHL-associated DNAm. In these SMR analyses, the mQTL summary data were collected from the study of Wu et al.[65], which perform the meta-analysis of Brisbane Systems Genetics Study and Lothian Birth Cohorts[27] (peripheral blood, $n = 1980$) and the eQTL summary data were obtained from the eQTLGen Consortium[26] (whole blood, $n = 14{,}115$). For visualization, we used OPERA[27] to generate the locus plot that integrated omics data, including co-localization patterns between genetic variants, molecular traits, and ARHL.

## Spatial distribution of ARHL related cells

To map the spatial distribution of cells associated with ARHL, we applied gsMap[29], a method based on the stratified LDSC, to integrate GWAS summary statistics with spatial transcriptomic (ST) data. For this analysis, we used mouse embryonic ST data from embryonic day 16.5 (E16.5) embryos, covering 18 organs obtained from the MOSTA database[30]. Because the ST data was derived from mouse tissue, the analysis was limited to homologous genes between human and mouse to ensure biological relevance. Before stratified LDSC (S-LDSC), gene specific scores (GSS) of each spot were assigned to nearby SNPs (located within a 50 kb window of the gene's transcription start site). These spot-specific SNP annotations were used to compute stratified LD scores based on a reference LD panel. We then applied S-LDSC to estimate the association between each spatial spot and ARHL risk. To assess the enrichment of ARHL heritability in specific tissues or cell-type regions, we used the Cauchy-combination test to aggregate $P$ values across spatial spots corresponding to each annotated tissue or cell type.

## Stratified LD score regression analysis for tissue, cell type specificity

To investigate which tissue and cell types contributed to ARHL, we applied LDSC-SEG, MEGMA and scDRS[32–34]. These approach estimates SNP heritability enrichment in genes with specific expression patterns across tissue or cell types. As the GTEx resource lacks expression data from human cochlear tissue, we instead utilized publicly available single-cell RNA sequencing (scRNA-seq) dataset from the mouse cochlear generated by Jean et al.[31], which includes a total of 85,167 cells from 36 cell types. In addition, we incorporated four publicly available scRNA-seq datasets used by Eshel et al, encompassing samples across a broad age range (postnatal day 15 (P15) to 15 months)[35–38]. The details of scRNA-seq datasets were as follows: expression matrix and cell annotations from Iyer et al. (P15), Eshel et al. (3 months) and Sun et al. (1, 2, 5, 12 and 15 months) were obtained from the gEAR platform[66]. For the dataset of Xu et al. (P28)[36], we downloaded the expression matrix from GEO (GSE202920) and conducted scRNA-seq analysis according to the procedures described in the original publication, using the provided code. For comparison with the study of Trpchevska et al., we incorporated two mouse cochlear scRNA-seq datasets from Milon et al. and Ranum et al.[45,46]. While the former captured the spiral ganglion region and the lateral wall/stria vascularis but lack hair cells and Deiters' cells, the latter was limited to IHCs, OHCs, and Deiters' cells.

We processed and calculated gene expression specificity following the procedures described by Trpchevska et al. We first excluded genes that were not expressed in any cell types. Next, we retained only the 1:1 orthologous gene between mouse and human, identified using biomaRt[67,68]. Gene expression was then normalized to transcripts per million (TPM), scaled to 1 TPM per cell type. Finally, gene expression specificity was calculated per gene per cell type as:

$$Specificity_{g,c} = \frac{TPM_{g,c}}{\sum_{i=1}^{n} TPM_{g,i}} \tag{3}$$

where $g$ denotes the gene and $c$ denotes the cell type. For each cell type, we selected top 10% genes that ranked by the quality-related expression values for each cell type and applied ±100 kb window around gene's transcription start site (TSS) to obtain a genome annotation. We then ran LDSC-SEG and MAGMA using these data to assess SNP heritability enrichment across cochlear cell types. In parallel, we reformatted the scRNA-seq data into the input required by scDRS and computed disease relevance scores per cell, which were then aggregated to the cell type level.

## Statistics & reproducibility

No statistical method was used to predetermine sample size, and no data were excluded from the analyses.

## Reporting summary

Further information on research design is available in the Nature Portfolio Reporting Summary linked to this article.

## Data availability

All GWAS summary statistics used for ARHL meta-analysis are available as below: the ARHL GWAS of East Asian from BBJ is publicly available at https://pheweb.jp/pheno/Hearing_Loss; the ARHL GWASs of East Asian, European, African and Admixed American from MVP are available via the dbGap study accession number phs002453; the ARHL GWAS of European from Trpchevska et al. is available at https://zenodo.org/records/5769707#.Ybm6v33MKhx. The summary statistic of the cross ancestry meta-analysis from this study is available at https://zenodo.org/records/17141085. The summary-level xQTL data used for SMR are available as follow: eQTL data from eQTLGen project are available at https://www.eqtlgen.org/cis-eqtls.html and the meta-analysis data of mQTL from LBC and BSGS are available at https://cnsgenomics.com/software/smr/#Download. The Roadmap Epigenomics Mapping Consortium epigenomic annotations data are available for download at http://compbio.mit.edu/roadmap. The 1000 Genome project of European reference data (phase 3) are available at https://ftp.1000genomes.ebi.ac.uk/vol1/ftp/phase3/. The spatial transcriptomics data of mouse embryos at E16.5 used for gsMap is available at https://db.cngb.org/stomics/mosta/download. The single-cell RNA-seq data of mouse cochlea for Jean et al., Iyer et al., Eshel et al., and Sun et al. is available from the dataset access via the gEAR portal (https://umgear.org/p?s=7fd80bf5, https://umgear.org/p?s=728a05e2, https://umgear.org/p?s=bb49463b, and https://umgear.org/p?s=653896d7). Source data are provided in this paper.

## Code availability

Code used in GWAS meta-analysis and results visualization are available at Github (https://github.com/Crazzy-Rabbit/project_hearing_loss/), which has been archived on Zenodo and assigned a DOI: (https://doi.org/10.5281/zenodo.17614049).

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

## Acknowledgements

This research was supported by the Fundamental Research Funds for the Central Universities (Y.W.), the 1·3·5 project for disciplines of excellence, West China Hospital, Sichuan University (ZYYC24006 to Y.W., ZYJC20002 to H.Y.), National Key Research and Development Program of China grant 82171836 (H.Y.). The research was supported by the National Supercomputing Center in Chengdu. The numerical calculations in this paper have been done on the Hefei Advanced Computing

Center. This study makes use of data from the UK Biobank (project ID: 151441). We acknowledge the use of Grammarly for assistance with grammar correction.

## Author contributions

Y.W. conceived and designed the experiment. Y.W. and H.Y. supervised the study. L.S. conducted all analyses with the assistance or guidance from Y.W., H.Y., Y.Z., J.L., K.G., W.L., and H.H. L.S. and Y.W. wrote the manuscript with the participation of all authors. All the authors approved the final version of the manuscript.

## Competing interests

The authors declare no competing interests.
