## [Peer Review File · Nature Communications]

Multi-ancestry GWAS of age-related hearing loss identifies 140 loci and key cellular mechanisms

Corresponding Author: Dr Yang Wu

Version 0:

Reviewer comments:

Reviewer #1

(Remarks to the Author)
Review

Shi et al. present a multi-ancestry GWAS meta-analysis of age-related hearing loss (ARHL) in 1.5 million individuals, identifying 98 independent risk loci, including 37 novel signals. The study integrates fine-mapping, genetic correlation, Mendelian randomization, and single-cell transcriptomic analyses to prioritize candidate genes, regulatory elements, and implicated cell types. The authors emphasize the identification of novel risk loci, the demonstration of negative selection acting on ARHL-associated variants, and the expansion of implicated cell types beyond previous studies. While the manuscript is comprehensive and employs state-of-the-art statistical approaches in GWAS, many of its findings overlap with the prior study by Trpchevska et al. (e.g., genetic correlations, missense variants), with incremental advances primarily being in the inclusion of additional ancestry groups, DNA methylation mapping, and cell-type enrichment analyses that identify new cell types involved in hearing loss.

While Trpchevska et al. demonstrated that cell in the stria vascularis are enriched with GWAS signals for hearing loss, Eshel et al. (<https://doi.org/10.1016/j.ajhg.2024.01.008>) suggest this is not the case, using other existing GWAS and scRNAseq resources. These contradicting findings are not solved in the current study, but the authors have the possibility of doing so.

Major Comments

1. The majority of loci and functional conclusions mirror those of Trpchevska et al. Of the 98 loci, many were previously reported, and the manuscript's main novelty lies in the identification of 37 new loci and refined fine-mapping. The meta-analysis includes Trpchevska et al., MVP, and Biobank Japan, but omits other large GWAS such as those by De Angelis (<https://doi.org/10.5281/zenodo.7897038>), Ivarsdottir (<http://www.decode.com/summarydata>), and Praveen (mainly UKBB driven), which are publicly available and would provide additional power and diversity (please see <https://doi.org/10.1124/pharmrev.124.001195> for some insights on the 4 studies). The authors do use de-meta-analysis to remove UK Biobank from their summary statistics before genetic correlation analyses, but should apply the same strategy to enable inclusion of the studies from De Angelis et al. and Ivarsdottir et al. in the main meta-analysis to maximize discovery and generalizability. Authors should include details on phenotype harmonization, ancestry representation, and technical compatibility (as evidenced in <https://doi.org/10.1124/pharmrev.124.001195>).
2. The use of SBayesRC for polygenic risk estimation and fine-mapping is appropriate and reflects current best practices. The application of SMR and HEIDI to integrate GWAS and eQTL/mQTL data is well executed but the tissue used (primarily blood) is not suitable for extrapolating on cochlear mechanisms. Despite the fact authors acknowledge this limitation, blood eQTL/mQTL analysis does not yield relevant findings. Authors should remove these results, and should mention the limitations of non-existing human cochlear eQTL resources in the discussion.
3. The LDSC-SEG approach is informative but incomplete. The authors expand the list of implicated cochlear cell types compared to Trpchevska et al. However, the interpretation of enrichment in the stria vascularis versus sensory epithelium remains unclear. Recent work by Eshel et al. argues that cochlear epithelial cells, not the stria vascularis, are the primary site of GWAS signal enrichment. There, Eshel et al. use a different scRNAseq resource and different GWAS studies but non systematically. Thus, the GWAS and scRNAseq dataset matter. Here is the opportunity to seal this debate with this 1.5

million sample size. First, authors should use both LDSC and Magma as Trpchevska et al., but also scDRS as used by Eshel et al. Only those cell types corroborated by at least two of these methods would be valid signals. Second, for replication purposes, authors should use the scRNAseq datasets that Trpchevska et al. have used, as well as that from Eshel et al. Since some of these signals were borderline significant, the new GWAS megasample, tested against the 3 scRNAseq datasets including Jean et al., would validate or not the suggestions from the past studies. Only adult mouse data should be used for comparison sake between the 3 scRNAseq studies. Ensure the method used is the exact same Trpchevska has reported, for sake of replication. The spatial transcriptomics at E16.5 from the mouse embryo is not relevant to humans with ARHL when compared to GTex – please remove, it does not add significantly novel information.

4. Of important note, the manuscript relies on the Jean et al. single-cell cochlear dataset, which, while comprehensive, uses 10X Genomics technology with limited sequencing depth. This restricts the ability to resolve rare or transcriptionally subtle cell types, such as subtypes of Type 1 auditory neurons, which have been missed by Jean et al. There were also batch differences in the Jean et al. study that are indicative of weaknesses in the validity of their sequencing. While Trpchevska et al. relied on the scRNAseq data from Millon et al., which also was performed on 10X genomics, the output successfully identified the 3 subtypes of Type 1 auditory neurons. This is why the corroboration, also through 3 distinct scRNAseq datasets, is needed. The authors should discuss the results from this analysis and the possibility that technical or reference biases that may influence the observed enrichments (which is why the use of the former datasets for replication and validation is needed).

5. Discuss the value of finding missense variations (found in familial cases, thus being rare), and identified here as common variants? Are these variants less penetrant in the studied population vs familial cases from more isolated consanguineous populations? What is the importance of such findings in a GWAS?

6. Authors should include sex stratified analyses as supplemental, with summary stats for the two sexes as well.

7. The absence of X-chromosome analysis should be discussed in the limitations – what would be expected from this analysis?

Minor comments:

The manuscript would benefit from more detailed descriptions of quality control, phenotype harmonization, and ancestry-specific analyses.

For loci claimed as novel, the authors should provide a comprehensive comparison with all recent GWAS, including those not included in the meta-analysis, to avoid overstatement of novelty.

The link to the summary statistics from the present study should be made available to the reviewers and opened to public once publication is accepted. <https://zenodo.org/records> is not sufficient for us to access the summary stats from the present meta-analysis. <https://umgear.org> is too vague – please specify the full link or provide more indications to access this dataset within umgear. Access to the codes: https://github.com/Crazy-Rabbit/project_hearing_loss/ was not possible. Please allow the reviewer to verify the codes.

Corrected FDR values for significance in Figure 6 seems wrong. Please check. Shouldn't it be: $p < 0,00139$?

I'd recommend one supplemental excel file with each supplementary table on separate sheets.

The use of AI for text and grammar should be acknowledged in the manuscript.

Reviewer #2

(Remarks to the Author)

In this study, Shi and colleagues conducted the largest cross-ancestry genome-wide association study (GWAS) to date on age-related hearing loss (ARHL), and applied a range of cutting-edged quantitative genetics approaches for post-GWAS analyses. There are some interesting results that will be of interest to the readership. However, I have a few comments below that I think could enhance the quality of the manuscript.

1. Please include Cochran's Q statistic or I² value for each SNP to quantify the heterogeneity in the cross-ancestry meta-analysis GWAS.
2. In addition to comparing effect sizes between EUR and non-EUR populations for the highlighted SNPs, it is also valuable to examine their MAF across different populations.
3. Page #6: What is the sample size of the de-meta-analysed GWAS after excluding the UKB cohort? The substantial reduction in power may make it less representative of the original cross-ancestry meta-analysis, potentially introducing bias into the BADGERS analysis. I recommend using Popcorn to estimate cross-population genetic correlations as an additional sensitivity check for BADGERS, which can account for potential sample overlap during estimation.
4. To the best of my knowledge, GSMR cannot reliably distinguish between correlated and uncorrelated pleiotropy from true causality, although it uses the HEIDI-outlier method to exclude instrumental SNPs that are likely pleiotropic to some extent. I recommend conducting additional sensitivity analyses (e.g., two sample MR, CAUSE, LHC-MR, etc) to re-assess the significant GSMR findings and help identify potential confounding effects due to pleiotropy.
5. For the results generated by S-LDSC, were they adjusted using the baseline model? If not, I recommend re-running the analysis with baseline model adjustment.

Minor:

6. Page #4: Please state how many independent SNP loci from the study are replicated compared to Trpchevska et al.
7. Page #4: Please provide the genetic correlation between the cross-ancestry meta-analysis GWAS and the FinnGen replication GWAS.
8. Page #6: Please remove the extra space in "Fig.4 a"
9. Some quantitative genetics methods (e.g., SMR, S-LDSC) appear to have limited ability to account for LD patterns across different populations, which may represent a limitation of these analyses.

Reviewer #3

(Remarks to the Author)

This manuscript presents a large-scale, multi-ancestry GWAS meta-analysis of age-related hearing loss incorporating ~1.5 million individuals. The authors identify 98 genome-wide significant loci (37 novel), perform fine-mapping and functional follow-up using multi-omics data, and explore cell-type heritability enrichment using spatial and single-cell transcriptomic data. The scale, multi-ancestry approach to GWAS, and integrative analyses are well thought-out and the paper is well-written. However, there are methodological concerns that should be addressed.

Major Comments

1. The manuscript defines a locus as novel if it is not previously reported and is located >500 kb from any known lead SNP. However, this approach risks both false positive (loci >500kb from previous hit that are actually in strong long-range LD) and false negatives (loci within 500kb but not in LD with known hits may be genuinely novel). It would strengthen the authors' argument to account for LD structure in addition to physical distance.
2. The replication sample is FinnGen. This is a genetically isolated European population. Given the stated emphasis on multi-ancestry discovery in this project, the use of a European-only replication cohort raises concerns about generalizability. Additionally, many downstream analyses use 1000 Genomes 3 European LD reference panels (e.g., LDSC, GSMR). This introduces potential biases when applied to the non-European ancestries represented in the discovery meta-analysis.
3. The authors examine genetic correlations with 1,738 traits in UK Biobank but do not explain how these traits were selected. Were they filtered for heritability, relevance, or redundancy?
4. The manuscript reports fine-mapping results using GWFM showing 1,042 variants in 157 credible sets and identifies only 26 SNPs with PIP >0.9. This resolution appears modest given the large sample size of the study. It would be beneficial to show fine-mapping diagnostics and compare to another rigorously benchmarked fine-mapping method like Susie to help contextualize performance.
5. The authors use ST data from mouse embryos to infer spatial enrichment of ARHL heritability, but ARHL is a late-onset disease. Further justification of this embryonic data as a suitable proxy for this human condition is needed.

Minor Comments

1. The LDSC intercept is reported (1.0095) and it suggests minimal inflation due to confounding. However, given the extremely large sample size of the analysis, it would be helpful to also report lambda 1000 (the genomic inflation factor standardized to 1000 cases/controls) to further evaluate inflation relative to scale.
2. The link to the GitHub code for this project is broken.
3. In supplemental table 5, there is a row with NaN values (noisy workplace). What does this indicate?
4. Please define beta hat in Figure 4b.

Version 1:

Reviewer comments:

Reviewer #1

(Remarks to the Author)

I thank the authors for the effort in making this analysis clear, as the one from Jean et al. is much more homogeneous and unified for the paper, than the several datasets used in Eshel et al.

The results using Jean et al. show that 2 out 3 enrichment methods show both hair cells, support cells, but also basal and root cells from the stria as major contributors to hearing loss.

As mentioned above in my former comment, 2 out 3 methods are sufficient for demonstrating a crucial role in hearing loss. If not all cells of the stria are implied, basal and root cells are consistently shown in the Jean et al. dataset. The authors find supportive evidence that both papers Trpchevska et al. and Eshel et al. that were conflicting each other, were correct. Please modify the statements in the manuscript accordingly, in order not to minimize the importance of basal and root cells from the stria in the pathogenesis of hearing loss.

As lines are not numbered, it is hard to point exactly to where the changes have to be done. Please check thoroughly.

--

(pages 11 in results and in discussion, and in abstract; page 13 "rather than the stria vascularis" – "However, this enrichment did not persist when larger-scale datasets with more comprehensive cell-type coverage were included"). It did persist in MAGMA and scDRS – this supports the role of the stria basal and root cells.

--

Finally, the lack of inclusion of two other GWAS datasets is a pity. I personally contacted the lead authors, but none responded. The icelandic dataset is indeed missing from the Decode database, and Regeneron only released fractionated summary statistics and not the pooled one. Very sad the journals and reviewers had missed this key information in these publications. The discussion needs to state somehow that attempts to gather the summary stats from these two other studies failed, which is an hindrance to the advances of science in general, and that it could have provided a complete analysis of the published literature. Yet, the compiled analysis from this study enables to demonstrate that the conflicting results from Trpchevska and Eshel are now solved.

Reviewer #2

(Remarks to the Author)

The authors fully addressed my previous concerns.

Reviewer #3

(Remarks to the Author)

The author has adequately addressed all my previous comments.

Version 2:

Reviewer comments:

Reviewer #1

(Remarks to the Author)

Thank you very much, the manuscript is a fantastic contribution to the field. Many thanks for the endeavors in addressing my comments. Congratulations!

Response to Reviewers' Comments

We thank the three reviewers for their time spent on reading our manuscript and for their constructive comments, which have helped us to improve the manuscript. We have responded to all the reviewers' comments point-by-point below (in blue) and have highlighted the relevant changes (in yellow) in the manuscript files.

Reviewer #1:

Shi et al. present a multi-ancestry GWAS meta-analysis of age-related hearing loss (ARHL) in 1.5 million individuals, identifying 98 independent risk loci, including 37 novel signals. The study integrates fine-mapping, genetic correlation, Mendelian randomization, and single-cell transcriptomic analyses to prioritize candidate genes, regulatory elements, and implicated cell types. The authors emphasize the identification of novel risk loci, the demonstration of negative selection acting on ARHL-associated variants, and the expansion of implicated cell types beyond previous studies.

While the manuscript is comprehensive and employs state-of-the-art statistical approaches in GWAS, many of its findings overlap with the prior study by Trpchevska et al. (e.g., genetic correlations, missense variants), with incremental advances primarily being in the inclusion of additional ancestry groups, DNA methylation mapping, and cell-type enrichment analyses that identify new cell types involved in hearing loss.

While Trpchevska et al. demonstrated that cell in the stria vascularis are enriched with GWAS signals for hearing loss, Eshel et al. (<https://doi.org/10.1016/j.ajhg.2024.01.008>) suggest this is not the case, using other existing GWAS and scRNAseq resources. These contradicting findings are not solved in the current study, but the authors have the possibility of doing so.

Major Comments

1. The majority of loci and functional conclusions mirror those of Trpchevska et al. Of the 98 loci, many were previously reported, and the manuscript's main novelty lies in the identification of 37 new loci and refined fine-mapping. The meta-analysis includes Trpchevska et al., MVP, and Biobank Japan, but omits other large GWAS such as those by De Angelis (<https://doi.org/10.5281/zenodo.7897038>), Ivarsdottir (<http://www.decode.com/summarydata>), and Praveen (mainly UKBB driven), which are publicly available and would provide additional power and diversity (please see <https://doi.org/10.1124/pharmrev.124.001195> for some insights on the 4 studies). The authors do use de-meta-analysis to remove UK Biobank from their summary statistics before genetic correlation analyses, but should apply the same strategy to enable inclusion of the studies from De Angelis et al. and Ivarsdottir et al. in the main meta-analysis to maximize discovery and generalizability. Authors should include details on

phenotype harmonization, ancestry representation, and technical compatibility (as evidenced in <https://doi.org/10.1124/pharmrev.124.001195>).

Re: We thank the reviewer for these constructive comments. Compared with the study by Trpchevska et al.¹, our work provides three key advances. First, whereas Trpchevska et al. focused only on European cohorts, we performed a cross-ancestry meta-analysis, enabling the assessment of cross-population similarities and improving the generalizability of findings. Second, by increasing the sample size from ~0.7 million to 1.5 million individuals, we substantially boosted discovery power and identified 140 independent loci, nearly a threefold increase in locus discovery. Third, our cell type enrichment analyses highlighted hair cells and supporting cells as the most strongly enriched cell types, in contrast to the stria vascularis reported by Trpchevska et al., but consistent with Eshel et al.². We found this discrepancy is likely attributable to limited cell-type coverage used in Trpchevska et al.¹ (see detailed response below). We have added these points in our revised Discussion section (page 11, 12).

As suggested by the reviewer, we have now updated our meta-analysis to incorporate the GWAS summary statistics from De Angelis et al.³ after removing the UK Biobank samples from this dataset (page 15). Unfortunately, the summary statistics from Ivarsdottir et al.⁴ are no longer accessible at the link provided, and those from Praveen et al.⁵ is not public available. We therefore can not include them in our meta-analysis. With the updated meta-analysis, we detected 140 independent SNPs, of which 44 were novel after comprehensive comparison with existing studies (Figure R1).

In addition, we have now provided details on the phenotype harmonization, ancestry representation and technical compatibility in Methods section (page 16, 17; Supplementary Table 1).

Figure R1 Manhattan plot of the cross-ancestry GWAS meta-analysis. The horizontal grey line denotes the genome-wide significance threshold ($P < 5 \times 10^{-8}$). In total, 44 novel independent risk loci are highlighted in red, with their nearest annotated genes shown in black text.

2. The use of SBayesRC for polygenic risk estimation and fine-mapping is appropriate and reflects current best practices. The application of SMR and HEIDI to integrate GWAS and eQTL/mQTL data is well executed but the tissue used (primarily blood) is not

suitable for extrapolating on cochlear mechanisms. Despite the fact authors acknowledge this limitation, blood eQTL/mQTL analysis does not yield relevant findings. Authors should remove these results, and should mention the limitations of non-existing human cochlear eQTL resources in the discussion.

Re: We agree with the reviewer that the lack of human cochlear eQTL/mQTL resources limits the mechanistic interpretation, and we have clarified this as a limitation in the revised discussion (page 14). Nevertheless, we note that the use of blood-derived eQTLs/mQTLs as a proxy tissue is a widely used practice when disease-relevant tissues are unavailable. For example, Jansen et al. used blood eQTLs to prioritize candidate genes for Alzheimer's disease, Han et al. integrated primary open-angle glaucoma (POAG) GWAS with blood pQTLs to identify putative casual proteins^{6,7}. Similar to these studies, we present our blood-based eQTL/mQTL analyses as preliminary findings, while highlighting the need for generating human cochlear multi-omics QTL datasets to validate and extend these results.

3. The LDSC-SEG approach is informative but incomplete. The authors expand the list of implicated cochlear cell types compared to Trpchevska et al. However, the interpretation of enrichment in the stria vascularis versus sensory epithelium remains unclear. Recent work by Eshel et al. argues that cochlear epithelial cells, not the stria vascularis, are the primary site of GWAS signal enrichment. There, Eshel et al. use a different scRNAseq resource and different GWAS studies but non systematically. Thus, the GWAS and scRNAseq dataset matter. Here is the opportunity to seal this debate with this 1.5 million sample size. First, authors should use both LDSC and Magma as Trpchevska et al., but also scDRS as used by Eshel et al. Only those cell types corroborated by at least two of these methods would be valid signals. Second, for replication purposes, authors should use the scRNAseq datasets that Trpchevska et al. have used, as well as that from Eshel et al. Since some of these signals were borderline significant, the new GWAS megasample, tested against the 3 scRNAseq datasets including Jean et al., would validate or not the suggestions from the past studies. Only adult mouse data should be used for comparison sake between the 3 scRNAseq studies. Ensure the method used is the exact same Trpchevska has reported, for sake of replication. The spatial transcriptomics at E16.5 from the mouse embryo is not relevant to humans with ARHL when compared to GTeX – please remove, it does not add significantly novel information.

Re: We thank the reviewer for this important and constructive comment. In response, we have substantially revised our cell type-specific enrichment analyses. Specifically, in addition to the dataset of Jean et al., we incorporated the additional scRNA-seq datasets used by Eshel et al. and Trpchevska et al., and systematically applied three different methods (LDSC-SEG, MAGMA, and scDRS) to integrate our GWAS with these scRNA-seq resources^{1,2}. For all analyses, we strictly followed the procedures reported by Trpchevska et al. to ensure methodological consistency and enable direct replication.

Across these analyses, we found that hair cells and supporting cells showed the most consistent and robust enrichment, suggesting that they are the primary contributors to the genetic basis of ARHL (Figure R2). This result is in agreement with Eshel et al., but different from those reported from Trpchevska et al., who emphasized the top enrichment in the stria vascularis. To investigate this discrepancy, we repeated heritability enrichment analysis using other two scRNA-seq datasets used in Trpchevska et al. (Milon et al. and Ranum et al.)^{8,9}, and successfully replicated the reported enrichment in the stria vascularis by Trpchevska et al. (Figure R3). However, this enrichment did not persist when larger-scale datasets with more comprehensive cell-type coverage were included. We found that this discrepancy is most likely explained by the limitation of cell type coverage in the datasets used by Trpchevska et al.^{8,9}. Specifically, cell-type specificity was computed as,

$$Specificity_{g,c} = \frac{TPM_{g,c}}{\sum_{i=1}^n TPM_{g,i}}$$

where g denotes a gene and c is a cell type. This metric quantifies, for each gene, how specific its expression is to a given cell type relative to all other cell types profiled. When the coverage of profiled cell types is limited, specificity scores can become inflated, leading to spurious enrichment. Our findings therefore underscore the importance of using datasets with comprehensive cell-type coverage to achieve reliable inference of cell-type-trait associations.

We have included these results and clarifications in revised Results and Discussions (page 10, 11, 13; Figure 6c; Supplementary Figure 14 and Figure 15; Supplementary Table 17-19).

Figure R2 Cell-type-specific heritability enrichment across three methods using comprehensive scRNA-seq datasets. Each row corresponds to one method. The first column shows results from the scRNA-seq dataset from Jean et al. (P8-20), and the remaining columns show results from the scRNA-seq dataset used by Eshel et al. Significance was controlled at FDR < 0.05 using the Benjamini-Hochberg (BH) procedure. The dashed line shows $-\log_{10}(P)$, where P is the dataset specific nominal threshold (i.e., the raw P of the least BH-significant test). If absent, no tests passed the BH-FDR < 0.05 threshold.

Figure R3 Cell-type-specific heritability enrichment across three methods using scRNA-seq dataset used by Trpchevska et al.. Each column corresponds to one method. The upper panel shows results based on the dataset from Milton et al. and lower panel shows results based on the dataset from Ranum et al.. Significance was controlled at $FDR < 0.05$ using the Benjamini-Hochberg (BH) procedure. The dashed line shows $-\log_{10}(P)$, where P is the dataset specific nominal threshold (raw P of the least BH-significant test). If absent, no tests passed the BH- $FDR < 0.05$ threshold.

For the used spatial transcriptomics dataset, we agree with the reviewer that embryonic mouse data are not ideal for studying ARHL. However, given the current lack of human embryonic datasets, a surrogate model is necessary. Prior studies (e.g., Breschi et al.¹⁰) have shown that mouse remains a reliable model with broadly conserved human-mouse gene expression profiles. Moreover, the gsMap method has successfully recapitulated known organ-trait associations¹¹ using mouse embryos at E16.5. Accordingly, we present these analyses as preliminary results. We acknowledge the limitations of relying on mouse embryonic data and highlight that human embryonic ST datasets are required to validate and extend these findings. We included this limitation in our revised Discussion section (page 14).

4. Of important note, the manuscript relies on the Jean et al. single-cell cochlear dataset, which, while comprehensive, uses 10X Genomics technology with limited sequencing depth. This restricts the ability to resolve rare or transcriptionally subtle cell types, such as subtypes of Type 1 auditory neurons, which have been missed by Jean et al. There were also batch to batch differences in the Jean et al. study that are indicative of weaknesses in the validity of their sequencing. While Trpchevska et al. relied on the scRNAseq data from Millon et al., which also was performed on 10X genomics, the output successfully identified the 3 subtypes of Type 1 auditory neurons. This is why the corroboration, also through 3 distinct scRNAseq datasets, is needed. The authors should discuss the results from this analysis and the possibility that technical or reference biases that may influence the observed enrichments (which is why the use of the former datasets for replication and validation is needed).

Re: We fully acknowledge the limitations of the Jean et al.¹² scRNA-seq dataset, including the shallow sequencing depth inherent to the 10X Genomics platform, which may hinder the detection of rare or transcriptionally subtle cell types populations such as subtype of Type I auditory neurons, as well as the reported batch-to-batch variation.

To address this concern, and in line with the reviewer's suggestion, we expanded our analyses to also incorporate the scRNA-seq datasets used by Trpchevska et al. and Eshel et al., in addition to Jean et al., thereby enabling replication and validation across three independent resources. We further applied three different enrichment methods (LDSC-SEG, MAGMA, scDRS) to strengthen the robustness of our findings. By corroborating results across multiple datasets and analytical methods, we were able to minimize dataset-specific biases and increase confidence in the enrichment results (see results showed in response above).

We have included in discussion (page 14) about how potential technical biases (e.g., sequencing depth, batch effects) and reference-based limitations could influence the observed enrichments, highlighting the importance of validating enrichment signals across diverse datasets.

5. Discuss the value of finding missense variations (found in familial cases, thus being rare), and identified here as common variants? Are these variants less penetrant in the studied population vs familial cases from more isolated consanguineous populations? What is the importance of such findings in a GWAS?

Re: We thank the reviewer for this comment. Several missense variants previously reported in familial hearing loss were also identified in our GWAS, consistent with evidence that 5%-10% of GWAS signals are coding variants. While these genes are primarily implicated in monogenic non-syndromic hearing loss, their reduced penetrance in the general population likely reflects the polygenic architecture of age-related hearing loss, as well as the influence of genetic modifiers and environmental factors.

We think these findings are valuable because they both provide independent support for genes previously implicated in familial hearing loss and highlight shared biological mechanisms between monogenic and polygenic forms of hearing loss. We have included this discussion in the revised Discussion (page 12).

6. Authors should include sex stratified analyses as supplemental, with summary stats for the two sexes as well.

Re: We thank the reviewer for this suggestion. Among all GWAS datasets included in our meta-analysis, only De Angelis et al. provide sex-stratified summary statistics. As other datasets lack results for sex-stratified analysis, we are unable to perform comprehensive sex-stratified meta-analyses. We have clarified this limitation in the revised Discussion section and highlighted the need for future studies with sex-stratified GWASs to better assess potential sex-specific effects (page 14).

7. The absence of X-chromosome analysis should be discussed in the limitations – what would be expected from this analysis?

Re: We thank the reviewer for this comment. We agree that X-chromosome analyses would be helpful to reveal sex-specific or dosage-related genetic effects not captured by autosomal variants¹³, and may provide additional biological insights. However, as most GWAS datasets included in our meta-analysis did not provide the X-chromosome specific results, we were unable to perform this analysis. We have acknowledged this as a limitation and added it to the revised Discussion (page 14).

Minor comments:

The manuscript would benefit from more detailed descriptions of quality control, phenotype harmonization, and ancestry-specific analyses.

Re: We agree with the reviewer and have now added more details on the quality control, phenotype harmonization and ancestry representation in the revised Methods (page 16, 17) and Supplementary Table 1.

For loci claimed as novel, the authors should provide a comprehensive comparison with all recent GWAS, including those not included in the meta-analysis, to avoid overstatement of novelty.

Re: We thank the reviewer for raising this point. In the revised analysis, we compared our lead SNPs with those reported by Ivarsdottir et al., Kalra et al., Praveen et al., De Angelis et al., Wells et al. and Trpchevska et al. ^{1,3,5,14,15}, to ensure a comprehensive assessment and avoid overstatement of novelty. Based on this comprehensive

comparison, we identified 44 novel loci not previously reported by these studies. We have included these clarifications in revised manuscript (page 18).

The link to the summary statistics from the present study should be made available to the reviewers and opened to public once publication is accepted.

<https://zenodo.org/records> is not sufficient for us to access the summary stats from the present meta-analysis. <https://umgear.org> is too vague – please specify the full link or provide more indications to access this dataset within umgear. Access to the codes: https://github.com/Crazy-Rabbit/project_hearing_loss/ was not possible. Please allow the reviewer to verify the codes.

Re: We apologize for incomplete or missing link. We have now made the GWAS summary statistic publicly available at <https://zenodo.org/records/17141085> and the single-cell RNA-seq data of mouse cochlea available at <https://umgear.org/p?s=7fd80bf5>. The GitHub repository with all analysis code is now publicly released at https://github.com/Crazy-Rabbit/project_hearing_loss/. These resources are all referenced in the Data Availability and Code Availability sections of our revised manuscript (page 23).

Corrected FDR values for significance in Figure 6 seems wrong. Please check. Shouldn't it be: $p < 0,00139$?

Re: In Figure 6, the reported values are FDR values calculated using the Benjamini-Hochberg procedure, with significance defined as $FDR < 0.05$. The threshold suggested by the reviewer ($P < 0.00139$) reflects a Bonferroni correction. While both approaches are commonly used to control multiple testing, we chose FDR as it provides greater discovery power. We have clarified this point and the definition of statistical significance in the revised manuscript and Figure 6 legend.

I'd recommend one supplemental excel file with each supplementary table on separate sheets.

Re: As suggested by the reviewer, we have now combined all supplementary tables into a single supplemental Excel file, with each table placed on a separate sheet.

The use of AI for text and grammar should be acknowledged in the manuscript.

Re: We have now added an acknowledgment in the revised manuscript that the AI tool Grammarly was used to assist with grammar correction.

Reviewer #2:

In this study, Shi and colleagues conducted the largest cross-ancestry genome-wide association study (GWAS) to date on age-related hearing loss (ARHL), and applied a range of cutting-edged quantitative genetics approaches for post-GWAS analyses. There are some interesting results that will be of interest to the readership. However, I have a few comments below that I think could enhance the quality of the manuscript.

1. Please include Cochran's Q statistic or I² value for each SNP to quantify the heterogeneity in the cross-ancestry meta-analysis GWAS.

Re: We thank the reviewer for this suggestion. In the revised manuscript, we have calculated Cochran's Q statistic and corresponding I² values for each SNP in the cross-ancestry meta-analysis. We have included these results in Supplementary Table 2.

2. In addition to comparing effect sizes between EUR and non-EUR populations for the highlighted SNPs, it is also valuable to examine their MAF across different populations.

Re: We thank the reviewer for this suggestion. In addition to comparing effect sizes between EUR and non-EUR populations, we have now examined the effect allele frequency (EAF) of the identified SNPs. The results showed a moderately high cross-ancestry concordance in EAF across ancestries (Figure R4). We have included this result in the revised Results section (page 5) and Supplementary Figure 4.

Figure R4 Pairwise comparisons of lead SNP effect allele frequency between European and three other ancestry groups, East Asian (a), African (b) and Admixed American (c). Each dot shows the effect size of a lead SNP, and the error bars show the 95% confidence interval of the estimated effect sizes. Abbreviations: EUR, European ancestry; EAS, East Asian ancestry; AMR, Admixed American ancestry; AFR, African ancestry.

3. Page #6: What is the sample size of the de-meta-analysed GWAS after excluding the UKB cohort? The substantial reduction in power may make it less representative of the original cross-ancestry meta-analysis, potentially introducing bias into the BADGERS analysis. I recommend using Popcorn to estimate cross-population genetic correlations as an additional sensitivity check for BADGERS, which can account for potential sample overlap during estimation.

Re: We thank the reviewer for this comment. After excluding the UK Biobank samples, the de-meta-analysis included 342,295 cases and 730,385 controls (total N = 1,072,680). As suggested by the reviewer, we used Popcorn to estimate cross-population genetic correlations, which yielded an estimate of 0.98 (se = 0.0026), indicating very high concordance between meta-analysis and de-meta-analysis. This result suggests that despite the reduction in sample size, the underlying genetic architecture before and after de-meta-analysis remain consistent, and the BADGERS results are less likely to be biased.

To further assess robustness, we applied LDSC to estimate the genetic correlations between our ARHL GWAS meta-analysis (without de-meta-analysis) and the 12 non-auditory phenotypes identified by BADGERS using the de-meta results. These correlations remained strong and statistically significant (Supplementary Table 5), further supporting the reliability of the BADGERS findings. We have added these results in our revised Results section (page 6, 7; Supplementary Table 5).

4. To the best of my knowledge, GSMR cannot reliably distinguish between correlated and uncorrelated pleiotropy from true causality, although it uses the HEIDI-outlier method to exclude instrumental SNPs that are likely pleiotropic to some extent. I recommend conducting additional sensitivity analyses (e.g., two sample MR, CAUSE, LHC-MR, etc) to re-assess the significant GSMR findings and help identify potential confounding effects due to pleiotropy.

Re: We agree with the reviewer that GSMR alone cannot fully distinguish correlated or uncorrelated pleiotropy from true causality. To address this, we performed additional sensitivity analyses using different MR approaches, including IVW, PRESSO, Robust, Simple Median and RAPS¹⁶⁻¹⁹. The results from these complementary methods were largely consistent with our GSMR findings, supporting the robustness of the causal inference (Figure R5). We have included these results in revised manuscript (page 7; Supplementary Figure 6; Supplementary Table 7).

Figure R5 Causal effect estimates from multiple Mendelian Randomization (MR) methods using ARHL as outcome. Each panel represents one exposure trait; the y-axis shows the MR methods used, and the x-axis shows the corresponding effect estimates with 95% confidence intervals for ARHL as the outcome. Statistical significance ($P < 0.05$) is indicated by a solid circle.

5. For the results generated by S-LDSC, were they adjusted using the baseline model? If not, I recommend re-running the analysis with baseline model adjustment.

Re: In our study, the S-LDSC results were indeed adjusted using the baseline model. In addition, we have performed complementary analyses using MAGMA and scDRS, which were all included in our revised manuscript to further support these functional enrichment findings.

Minor:

6. Page #4: Please state how many independent SNP loci from the study are replicated compared to Trpchevska et al.

Re: In our updated meta-analysis, which additionally incorporated the GWAS summary statistic from De Angelis et al³, we identified 140 lead SNPs after LD clumping. Of these, 102 SNPs present in the GWAS of Trpchevska et al. Among them, 45 SNPs (44.12%) reached genome-wide significance ($P < 5 \times 10^{-8}$), and all of them reached the Bonferroni corrected significance ($P < 0.00049$). We have now included these results in revised Results section (page 4, 5).

7. Page #4: Please provide the genetic correlation between the cross-ancestry meta-analysis GWAS and the FinnGen replication GWAS.

Re: We thank the reviewer for this comment. As suggested by the reviewer, we calculated the genetic correlation between the cross-ancestry meta-analysis GWAS and the FinnGen replication GWAS using Popcorn. The estimated genetic correlation was 0.63 (se = 0.12), indicating a moderate level of concordance. We have included these results to the Results section (page 4).

8. Page #6: Please remove the extra space in “Fig.4 a”

Re: We thank the reviewer for pointing this error out. The extra space in “Fig. 4a” has been removed in the revised manuscript (page 6).

9. Some quantitative genetics methods (e.g., SMR, S-LDSC) appear to have limited ability to account for LD patterns across different populations, which may represent a limitation of these analyses.

Re: We agree that methods such as SMR and S-LDSC have limited ability to fully account for LD differences across populations. We have now added a statement in the revised Discussion to acknowledge this limitation (page 14).

Reviewer #3:

This manuscript presents a large-scale, multi-ancestry GWAS meta-analysis of age-related hearing loss incorporating ~1.5 million individuals. The authors identify 98 genome-wide significant loci (37 novel), perform fine-mapping and functional follow-up using multi-omics data, and explore cell-type heritability enrichment using spatial and single-cell transcriptomic data. The scale, multi-ancestry approach to GWAS, and integrative analyses are well thought-out and the paper is well-written. However, there are methodological concerns that should be addressed.

Major Comments

1. The manuscript defines a locus as novel if it is not previously reported and is located >500 kb from any known lead SNP. However, this approach risks both false positive (loci >500kb from previous hit that are actually in strong long-range LD) and false negatives (loci within 500kb but not in LD with known hits may be genuinely novel). It would strengthen the authors' argument to account for LD structure in addition to physical distance.

Re: We thank the reviewer for this constructive suggestion. In the revised analysis, we re-defined novel loci by considering both physical distance and LD structure.

Specifically, we classified a SNP as novel only if it was located more than 500kb away from any previously reported variants with $r^2 < 0.1$, or if it was within 500 kb of known loci but showed stronger LD independence ($r^2 < 0.05$). We have clarified this updated definition in the revised Methods section (page 18).

2. The replication sample is FinnGen. This is a genetically isolated European population. Given the stated emphasis on multi-ancestry discovery in this project, the use of a European-only replication cohort raises concerns about generalizability. Additionally, many downstream analyses use 1000 Genomes 3 European LD reference panels (e.g., LDSC, GSMR). This introduces potential biases when applied to the non-European ancestries represented in the discovery meta-analysis.

Re: We thank the reviewer for this comment. Although our study incorporates GWAS data from multiple ancestries, individuals of European ancestry account for ~ 77% of the total sample (Figure R6). Given this predominant European contribution, and the lack of large-scale cross-ancestry replication cohorts and ancestry-specific LD reference panels, we used FinnGen for replication and the 1000 Genomes European samples as LD reference for downstream analyses. We note that using a European-only replication cohort may result in a downward bias in replication rates, rather than inflated estimates. This strategy is consistent with those commonly used in cross-ancestry GWAS for other complex traits^{20,21}. We acknowledge that this approach may introduce biases and limit generalizability, and have added this limitation in the Discussion section (page 13, 14).

Figure R6 Overview of GWAS summary statistics used in the cross-ancestry meta-analysis of ARHL. The figure summarizes the contribution cohorts, ancestry composition, and sample sizes for each dataset included.

3. The authors examine genetic correlations with 1,738 traits in UK Biobank but do not explain how these traits were selected. Were they filtered for heritability, relevance, or redundancy?

Re: The 1,738 traits included in our genetic correlation analysis were selected from all UK Biobank traits based on nominally significant heritability ($P < 0.05$) estimated using LDSC, as described in BADGERS²². We have now included this clarification in the revised Methods section (page 19).

4. The manuscript reports fine-mapping results using GWFM showing 1,042 variants in 157 credible sets and identifies only 26 SNPs with PIP > 0.9. This resolution appears modest given the large sample size of the study. It would be beneficial to show fine-mapping diagnostics and compare to another rigorously benchmarked fine-mapping method like Susie to help contextualize performance.

Re: We thank the reviewer for this suggestion. In our updated meta-analysis with including additional GWAS summary data³, we identified 1,108 SNPs in 165 credible sets and identifies 22 SNPs with PIP > 0.9 after GWFM. We agree that the resolution may appear modest compared to other complex traits with similar sample sizes. However, this is likely due to the highly polygenic architecture of ARHL, which limits fine-mapping power.

To evaluate the robustness of our fine-mapping results, we re-analyzed all loci using the SuSiE method with the same GWAS summary statistics and LD reference. In the diagnostic analysis, SuSiE estimated a mean lambda of 0.006 across loci (Figure R7), indicating good concordance between the LD reference and our ARHL GWAS data. Further fine-mapping analysis with SuSiE identified 9 SNPs with PIP > 0.9, of which 7 overlapped with the 22 SNPs identified by our GWFM analysis. Among the 22 fine-mapped variants identified by GWFM, 7 achieved a SuSiE PIP > 0.9, and 50% had PIPs > 0.5 and 87% had PIPs > 0.1, confirming the consistency and reliability of our findings. We have added these results in our revised Results section (page 8; Supplementary Figure 10; Supplementary Table 11).

Figure R7 Concordance between the LD reference and ARHL GWAS. The histogram shows the distribution of λ values estimated from SuSiE diagnostics.

5. The authors use ST data from mouse embryos to infer spatial enrichment of ARHL heritability, but ARHL is a late-onset disease. Further justification of this embryonic data as a suitable proxy for this human condition is needed.

Re: We thank the reviewer for this comment. We agree that embryonic mouse spatial transcriptomics (ST) data are not ideal for studying ARHL. However, in the absence of adult human ST datasets, a proxy model is necessary. Similar strategies have been used for other late-onset complex traits, such as male pattern baldness (MPB), where mouse embryonic ST data revealed risk gene enrichment in facial epithelial cells that spatially cluster during the formation of hair follicles and express canonical follicle marker genes (*KRT15*, *KRT5*, and *KRT17*)¹¹. This example illustrates that embryonic ST data can provide valuable insights into the developmental origins of late-onset diseases. Nevertheless, we acknowledge the limitations of using mouse embryonic data as a proxy for ARHL and emphasize the need of generating adult human ST datasets, particularly inner-ear ST datasets, to validate and extend these findings. We have included this clarification in our revised Discussion section (page 13, 14).

Minor Comments

1. The LDSC intercept is reported (1.0095) and it suggests minimal inflation due to confounding. However, given the extremely large sample size of the analysis, it would be helpful to also report lambda 1000 (the genomic inflation factor standardized to 1000 cases/controls) to further evaluate inflation relative to scale.

Re: As suggested by the reviewer, in addition to reporting the LDSC intercept, we have now calculated lambda 1000, which was estimated to be 1.0006. This value further confirms that there is minimal inflation due to confounding in our GWAS meta-analysis. We have included this result in our revised Results section (page 4).

2. The link to the GitHub code for this project is broken.

Re: We thank the reviewer for pointing this error out. The GitHub code is now publicly available at https://github.com/Crazy-Rabbit/project_hearing_loss/. We have also updated this in the Code Availability section of our revised manuscript (page 23).

3. In supplemental table 5, there is a row with NaN values (noisy workplace). What does this indicate?

Re: The NaN values in Supplementary Table 5 indicate that the GSMR analysis could not be performed due to an insufficient number of valid SNP instruments (fewer than the required 10 SNPs after the clumping with $P < 5 \times 10^{-8}$ and LD $r^2 < 0.05$), resulting in no estimated values. We have clarified this in the revised table legend (Supplementary Table 6).

4. Please define beta hat in Figure 4b.

Re: We thank the reviewer for pointing out the missing definition. In Figure 4b, \hat{b}_{xy} denotes the estimated effect size from exposures (y-axis) on outcome (ARHL) obtained from the GSMR analysis. We have added this definition to the revised figure legend for clarity (page 32).

References

1. Trpchevska, N. *et al.* Genome-wide association meta-analysis identifies 48 risk variants and highlights the role of the stria vascularis in hearing loss. *The American Journal of Human Genetics* **109**, 1077–1091 (2022).
2. Eshel, M., Milon, B., Hertzano, R. & Elkon, R. The cells of the sensory epithelium, and not the stria vascularis, are the main cochlear cells related to the genetic pathogenesis of age-related hearing loss. *The American Journal of Human Genetics* **111**, 614–617 (2024).
3. De Angelis, F. *et al.* Sex differences in the polygenic architecture of hearing problems in adults. *Genome Medicine* **15**, 36 (2023).
4. Ivarsdottir, E. V. *et al.* The genetic architecture of age-related hearing impairment revealed by genome-wide association analysis. *Commun Biol* **4**, 706 (2021).
5. Praveen, K. *et al.* Population-scale analysis of common and rare genetic variation associated with hearing loss in adults. *Commun Biol* **5**, 540 (2022).
6. Jansen, I. E. *et al.* Genome-wide meta-analysis identifies new loci and functional pathways influencing Alzheimer's disease risk. *Nat Genet* **51**, 404–413 (2019).
7. Han, X. *et al.* Large-scale multitrait genome-wide association analyses identify hundreds of glaucoma risk loci. *Nat Genet* **55**, 1116–1125 (2023).
8. Milon, B. *et al.* A cell-type-specific atlas of the inner ear transcriptional response to acoustic trauma. *Cell Rep* **36**, 109758 (2021).
9. Ranum, P. T. *et al.* Insights into the Biology of Hearing and Deafness Revealed by Single-Cell RNA Sequencing. *Cell Rep* **26**, 3160–3171.e3 (2019).
10. Breschi, A., Gingeras, T. R. & Guigó, R. Comparative transcriptomics in human and mouse. *Nat Rev Genet* **18**, 425–440 (2017).
11. Song, L., Chen, W., Hou, J., Guo, M. & Yang, J. Spatially resolved mapping of cells associated with human complex traits. *Nature* **641**, 932–941 (2025).
12. Jean, P. *et al.* Single-cell transcriptomic profiling of the mouse cochlea: An atlas for targeted therapies. *Proceedings of the National Academy of Sciences* **120**, e2221744120 (2023).
13. Gorlov, I. P. & Amos, C. I. Why does the X chromosome lag behind autosomes in GWAS findings? *PLoS Genet* **19**, e1010472 (2023).
14. Ivarsdottir, E. V. *et al.* The genetic architecture of age-related hearing impairment revealed by genome-wide association analysis. *Commun Biol* **4**, 1–13 (2021).
15. Kalra, G. *et al.* Biological insights from multi-omic analysis of 31 genomic risk loci for adult hearing difficulty. *PLoS Genet* **16**, e1009025 (2020).
16. Burgess, S., Butterworth, A. & Thompson, S. G. Mendelian randomization analysis with multiple genetic variants using summarized data. *Genet Epidemiol* **37**, 658–665 (2013).
17. Verbanck, M., Chen, C.-Y., Neale, B. & Do, R. Detection of widespread horizontal pleiotropy in causal relationships inferred from Mendelian randomization between complex traits and diseases. *Nat Genet* **50**, 693–698 (2018).
18. Bowden, J., Davey Smith, G., Haycock, P. C. & Burgess, S. Consistent Estimation in Mendelian Randomization with Some Invalid Instruments Using a Weighted Median Estimator. *Genet Epidemiol* **40**, 304–314 (2016).

19. Zhao, Q., Wang, J., Hemani, G., Bowden, J. & Small, D. S. Statistical inference in two-sample summary-data Mendelian randomization using robust adjusted profile score. *The Annals of Statistics* **48**, 1742–1769 (2020).
20. Han, X. *et al.* Large-scale multitrait genome-wide association analyses identify hundreds of glaucoma risk loci. *Nat Genet* **55**, 1116–1125 (2023).
21. Major Depressive Disorder Working Group of the Psychiatric Genomics Consortium. Electronic address: andrew.mcintosh@ed.ac.uk & Major Depressive Disorder Working Group of the Psychiatric Genomics Consortium. Trans-ancestry genome-wide study of depression identifies 697 associations implicating cell types and pharmacotherapies. *Cell* **188**, 640-652.e9 (2025).
22. Yan, D. *et al.* BADGERS: biobank wide trait association. *eLife* **12**, RP91360 (2024).

Response to Reviewers' Comments

We thank all three reviewers for their positive comments on our revised manuscript. In this revision, we have responded to the additional comments from reviewer #1 point-by-point below (in blue) and have highlighted the relevant changes (in yellow) in the manuscript files.

Reviewer #1:

I thank the authors for the effort in making this analysis clear, as the one from Jean et al. is much more homogeneous and unified for the paper, than the several datasets used in Eshel et al.

Re: We thank the reviewer for acknowledging the improvements made in our revised manuscript. We appreciate the additional comments and have made changes below.

The results using Jean et al. show that 2 out of 3 enrichment methods show both hair cells, support cells, but also basal and root cells from the stria as major contributors to hearing loss.

Re: We thank the reviewer for this insightful comment. We have revised the text to explicitly state that, in the Jean et al. dataset, two out of three enrichment methods (MAGMA and scDRS) consistently identified not only hair cells and supporting cells but also the basal and root cells of the stria vascularis as important contributors to hearing loss. We have clarified this in the revised **Results** (page 10): "Across these analyses, hair cells and supporting cells consistently showed the strongest enrichment, suggesting they are the primary contributors to the genetic architecture of ARHL (Fig. 6c; Supplementary Table 17). Moreover, significant enrichment signals were also observed in the basal and root cell compartments of the stria vascularis in both MAGMA and scDRS analyses, highlighting a potential complementary role of these non-sensory epithelial cells in hearing loss pathogenesis."

As mentioned above in my former comment, 2 out of 3 methods are sufficient for demonstrating a crucial role in hearing loss. If not all cells of the stria are implied, basal and root cells are consistently shown in the Jean et al. dataset. The authors find supportive evidence that both papers Trpchevska et al. and Eshel et al. that were conflicting each other, were correct. Please modify the statements in the manuscript accordingly, in order

not to minimize the importance of basal and root cells from the stria in the pathogenesis of hearing loss. As lines are not numbered, it is hard to point exactly to where the changes have to be done. Please check thoroughly.

--

(pages 11 in results and in discussion, and in abstract; page 13 “rather than the stria vascularis” – “However, this enrichment did not persist when larger-scale datasets with more comprehensive cell-type coverage were included”). It did persist in MAGMA and scDRS – this supports the role of the stria basal and root cells.

--

Re: We thank the reviewer for raising this important point. We have checked the whole manuscript thoroughly and revised accordingly (including **Abstract**, **Results** and **Discussion**) to reflect that basal and root cells of the stria vascularis consistently show enrichment in the Jean et al. dataset and to ensure that their contribution is not missed. Our changes included:

Abstract (page 2): “Moreover, analyses incorporating spatial and single-cell transcriptomic identified the inner ear as a crucial site of ARHL, emphasizing the importance of hair cells, supporting cells, basal and root cells of the stria vascularis to its pathogenesis.”

Results (page 11): “These findings reconcile the apparent discrepancy between Eshel et al., who emphasized hair cells and supporting cells, and Trpchevska et al., who highlighted basal and root cells of the stria vascularis, by revealing their shared contribution to hearing loss. Collectively, these results provide robust evidence that ARHL heritability is significantly enriched in hair cells, supporting cells, and basal and root cells of the stria vascularis, reinforcing their collective roles in hearing loss pathogenesis.”

Discussion (page 11): “Third, our cell type enrichment analyses highlighted hair cells and supporting cells as the most strongly enriched cell types, while also revealing significant enrichment in basal and root cells of the stria vascularis, thereby reconciling the findings of Eshel et al. and Trpchevska et al.”

Discussion (page 13): “These findings reconcile the previously conflicting results of Eshel et al. and Trpchevska et al., providing a unified view of the cellular basis of ARHL heritability.”

Discussion (page 13): “When large-scale datasets with broader cell-type coverage were analyzed, enrichment was observed not only in basal and root cells of the stria vascularis but also in hair cells and supporting cells, indicating that the differences between studies likely result from variation in dataset composition rather than conflicting biological interpretations.”

Finally, the lack of inclusion of two other GWAS datasets is a pity. I personally contacted the lead authors, but none responded. The icelandic dataset is indeed missing from the Decode database, and Regeneron only released fractionated summary statistics and not the pooled one. Very sad the journals and reviewers had missed this key information in these publications. The discussion needs to state somehow that attempts to gather the summary stats from these two other studies failed, which is an hindrance to the advances of science in general, and that it could have provided a complete analysis of the published literature. Yet, the compiled analysis from this study enables to demonstrate that the conflicting results from Trpchevska and Eshel are now solved.

Re: We thank the reviewer for this comment. In response, we have added a statement in the **Discussion** explicitly describing our efforts to obtain these two additional GWAS datasets, which were not successful. We have clarified this in **Discussion** (page 15): “Finally, although we collected almost all available GWAS summary statistics for ARHL, the dataset from Ivarsdottir et al.¹ is currently missing from the Decode database, and that from Praveen et al.² is not publicly accessible. This lack of data accessibility limited our ability to perform a fully comprehensive meta-analysis, highlighting the need for broader data sharing to accelerate progress in the field.”

Reviewer #2:

The authors fully addressed my previous concerns.

Re: We thank the reviewer for the comments on all versions of our manuscript.

Reviewer #3:

The author has adequately addressed all my previous comments.

Re: We thank the reviewer again for the time taken to read our revised manuscript and for providing critical and constructive comments.

Reference

1. Ivarsdottir, E. V. *et al.* The genetic architecture of age-related hearing impairment revealed by genome-wide association analysis. *Commun Biol* **4**, 706 (2021).
2. Praveen, K. *et al.* Population-scale analysis of common and rare genetic variation associated with hearing loss in adults. *Commun Biol* **5**, 540 (2022).